# Exposure to air pollutants and subclinical carotid atherosclerosis measured by magnetic resonance imaging: A cross-sectional analysis

Sandi M. Azab[1,2], Dany Doiron[3], Karleen M. Schulze[2,4], Jeffrey R. Brook[5], Michael Brauer[6], Eric E. Smith[7], Alan R. Moody[8], Dipika Desai[4], Matthias G. Friedrich[9], Shrikant I. Bangdiwala[1,4], Dena Zeraatkar[10], Douglas Lee[11], Trevor J. B. Dummer[6], Paul Poirier[12], Jean-Claude Tardif [13], Koon K. Teo[2,4], Scott Lear[14], Salim Yusuf[4], Sonia S. Anand[1,2,4], Russell J. de Souza [1,4]*, for the Canadian Alliance of Healthy Hearts and Minds (CAHHM) Study Investigators[¶]

1 Department of Health Research Methods, Evidence, and Impact, McMaster University, Hamilton, Ontario, Canada, 2 Department of Medicine, McMaster University, Hamilton, Ontario, Canada, 3 Research Institute of McGill University Health Centre, Montreal, Canada, 4 Population Health Research Institute, Hamilton, Ontario, Canada, 5 Occupational and Environmental Health Division, Dalla Lana School of Public Health, University of Toronto, Toronto, Ontario, Canada, 6 School of Population and Public Health, The University of British Columbia, Vancouver, British Columbia, Canada, 7 Department of Clinical Neurosciences, University of Calgary, Calgary, Alberta, 8 Department of Medical Imaging, Sunnybrook Health Sciences Centre, University of Toronto, Toronto, Ontario, Canada, 9 Department of Medicine and Diagnostic Radiology, McGill University, Montreal, Quebec, Canada, 10 Department of Biomedical Informatics, Harvard Medical School, Boston, Massachusetts, United States of America, 11 Programming and Biostatistics, Institute for Clinical Evaluative Sciences, Toronto, Ontario, Canada, 12 Faculté de Pharmacie, Institut Universitaire de Cardiologie et de Pneumologie de Québec, Quebec City, Quebec, Canada, 13 Montreal Heart Institute, Université de Montréal, Montreal, Quebec, Canada, 14 Faculty of Health Sciences, Simon Fraser University, Burnaby, British Columbia, Canada

¶ The full list of CAHHM Study Investigators can be found under Acknowledgments.
* desouzrj@mcmaster.ca

**Data Availability Statement:** CAHHM data cannot be deposited publicly as these collaborative data originate from multiple Canadian cohorts with

## Abstract

### Objectives

Long-term exposure to air pollution has been associated with higher risk of cardiovascular mortality. Less is known about the association of air pollution with initial development of cardiovascular disease. Herein, the association between low-level exposure to air pollutants and subclinical carotid atherosclerosis in adults without known clinical cardiovascular disease was investigated.

### Design

Cross-sectional analysis within a prospective cohort study.

### Setting

The Canadian Alliance for Healthy Hearts and Minds Cohort Study; a pan-Canadian cohort of cohorts.

different legal frameworks. CAHHM represents a "cohort of cohorts", and at the time participants were enrolled into each respective cohort, data sharing was not part of the consent process. Thus, the appropriate legal frameworks to allow for data to be deposited publicly (identified or de-identified) is not in place, nor was such sharing anticipated or run by the respective REB at the time of cohort invitations. Therefore, any requests for data may be made to CAHHM (Alliance@phri.ca), with requests for data sharing to be considered on a case-by-case basis. A detailed statistical analysis plan and the code used in the analysis are also available on reasonable request.

**Funding:** CAHHM was funded by the Canadian Partnership Against Cancer (CPAC), Heart and Stroke Foundation of Canada (HSF-Canada), and the Canadian Institutes of Health Research (CIHR). Financial contributions were also received from the Population Health Research Institute and CIHR Foundation Grant no. FDN-143255 to S.S.A.; FDN-143313 to J.V.T.; and FDN 154317 to E.E.S. In-kind contributions from A.R.M. and S.E.B. from Sunnybrook Hospital, Toronto for MRI reading costs, and Bayer AG for provision of IV contrast. The Canadian Partnership for Tomorrow's Health is funded by the Canadian Partnership Against Cancer and Health Canada, BC Cancer, Genome Quebec, Centre Hospitalier Universitaire (CHU) Sainte-Justine, Dalhousie University, Ontario Institute for Cancer Research, Alberta Health, Alberta Cancer Foundation, and Alberta Health Services. The PURE Study was funded by multiple sources. The Montreal Heart Institute Biobank is funded by Mr André Desmarais and Mrs France Chrétien-Desmarais and the Montreal Heart Institute Foundation. The funders had no role in study design, data collection and analysis, decision to publish, or preparation of the manuscript.

**Competing interests:** I have read the journal's policy and the authors of this manuscript have the following competing interests: RJ de Souza has served as an external resource person to the World Health Organization's Nutrition Guidelines Advisory Group on trans fats, saturated fats, and polyunsaturated fats. The WHO paid for his travel and accommodation to attend meetings from 2012-2017 to present and discuss this work. He has presented updates of this work to the WHO in 2022. He has also done contract research for the Canadian Institutes of Health Research's Institute of Nutrition, Metabolism, and Diabetes, Health Canada, and the World Health Organization for which he received remuneration. He has received speaker's fees from the University of Toronto, and McMaster Children's Hospital. He has held grants

## Participants

Canadian adults (n = 6645) recruited between 2014–2018 from the provinces of British Columbia, Alberta, Ontario, Quebec, and Nova Scotia, were studied, for whom averages of exposures to nitrogen dioxide ($NO_2$), ozone ($O_3$), and fine particulate matter ($PM_{2.5}$) were estimated for the years 2008–2012.

## Main outcome measure

Carotid vessel wall volume (CWV) measured by magnetic resonance imaging (MRI).

## Results

In adjusted linear mixed models, $PM_{2.5}$ was not consistently associated with CWV (per 5 µg/m$^3$ $PM_{2.5}$; adjusted estimate = -8.4 mm$^3$; 95% Confidence Intervals (CI) -23.3 to 6.48; p = 0.27). A 5 ppb higher $NO_2$ concentration was associated with 11.8 mm$^3$ lower CWV (95% CI -16.2 to -7.31; p<0.0001). A 3 ppb increase in $O_3$ was associated with 9.34 mm$^3$ higher CWV (95% CI 4.75 to 13.92; p<0.0001). However, the coarse/insufficient $O_3$ resolution (10 km) is a limitation.

## Conclusions

In a cohort of healthy Canadian adults there was no consistent association between $PM_{2.5}$ or $NO_2$ and increased CWV as a measure of subclinical atherosclerosis by MRI. The reasons for these inconsistent associations warrant further study.

## Introduction

Cardiovascular disease (CVD) is a leading cause of mortality in Canada and worldwide, and traditional risk factors include smoking, obesity, diabetes, hypertension, and dyslipidemia [1]. Growing epidemiological evidence describes the adverse effects of air pollution on cardiovascular health [2–5], however the role of low levels of exposure to air pollution on early subclinical markers of cardiovascular dysfunction, *e.g.*, atherosclerosis, is not well-characterized [6].

Major pollutants include particulate matter (PM) and gaseous air pollutants, such as nitrogen oxides ($NO_x$) and ground level ozone ($O_3$) [7]. Fine particulate matter ($PM_{2.5}$) is fine inhalable particles $\leq$2.5 µm in aerodynamic diameter. Nitrogen dioxide ($NO_2$) is mainly associated with road traffic and other forms of fossil combustion and is a precursor to $O_3$, which is formed through chemical reactions between NOx and volatile organic compounds (VOC) in the presence of sunlight [8]. The 2022 special report by the Health Effects Institute (HEI) on long-term exposure to traffic-related air pollution (TRAP), which included 57 studies investigating cardiometabolic effects found that the confidence in the evidence for the association of air pollution with cardiovascular (circulatory and ischemic heart disease) mortality is high, but with cardiovascular morbidity is at best moderate to low [9, 10]. However, the methods of outcome assessment varied substantially and only half of the studies entered a meta-analysis of which one-third rated as high risk of bias for the confounder domain [9, 10].

Imaging of the carotid arteries is a non-invasive biomarker of subclinical atherosclerosis -the progressive buildup of plaques- for early prediction of IHD risk in healthy individuals without clinically significant CVD [11]. In the longitudinal Multiethnic Study of

from the Canadian Institutes of Health Research, Canadian Foundation for Dietetic Research, Population Health Research Institute, and Hamilton Health Sciences Corporation as a principal investigator, and is a co-investigator on several funded team grants from the Canadian Institutes of Health Research. He has served as an independent director of the Helderleigh Foundation (Canada). He serves as a member of the Nutrition Science Advisory Committee to Health Canada (Government of Canada), and a co-opted member of the Scientific Advisory Committee on Nutrition (SACN) Subgroup on the Framework for the Evaluation of Evidence (Public Health England). Dr Anand reported receiving grants from Canadian Partnership Against Cancer, Heart and Stroke Foundation of Canada, and Canadian Institutes of Health Research, and a Canadian Institutes of Health Research Foundation grant during the conduct of the study and serving as the Tier 1 Canada Research Chair Ethnicity and Cardiovascular Disease and as the Michael G Degroote Heart and Stroke Foundation Chair in Population Health Research, and receiving grants from Heart and Stroke Foundation of Canada and Canadian Institutes of Health Research, and receiving personal fees from Bayer outside the submitted work. Dr Friedrich reported receiving personal fees from Circle CVI Inc for serving as a board member and adviser and being a shareholder outside the submitted work. Dr Dummer reported receiving grants from Canadian Partnership Against Cancer during the conduct of the study. Dr Lear reported receiving grants from the Canadian Institutes of Health Research and grants from Michael Smith Foundation for Health Research during the conduct of the study and personal fees from Curatio Inc outside the submitted work. Dr Tardif reported receiving grants from Amarin, Ceapro, Esperion, Ionis, Novartis, Pfizer, RegenXBio, Sanofi, AstraZeneca, and DalCor Pharmaceuticals, receiving personal fees from AstraZeneca, HLS Pharmaceuticals, Pendopharm, and DalCor Pharmaceuticals, and having a minor equity interest in DalCor Pharmaceuticals Minor outside the submitted work. In addition, Dr Tardif had a patent for Pharmacogenomics-Guided CETP Inhibition issued by DalCor Pharmaceuticals, a patent for Use of Colchicine After Myocardial Infarction pending, and a patent for Genetic Determinants of Response to Colchicine pending. No other disclosures were reported. Dr Brauer served on the WHO Guideline Development Group (no remuneration was provided but travel costs to meetings were covered). This does not alter our adherence to PLOS ONE policies on sharing data and materials.

Atherosclerosis (MESA), $PM_{2.5}$, $NO_2$, and $NO_X$ were not associated with carotid intima-media thickness (cIMT) change, while ambient $O_3$ was associated with increased progression of cIMT [2, 8]. Magnetic resonance imaging (MRI) can accurately assess the presence of subclinical cerebrovascular atherosclerosis [11] by measuring carotid vessel wall volume (CWV) i.e. the entire thickness of the wall. MRI-determined CWV, compared to ultrasound-measured cIMT, includes the adventitia, which is the source of vasa vasorum that further proliferates with arterial wall thickening [12]. Thus, CWV is a more sensitive measure of early plaque development [12, 13] and more consistently associated with incident CVD than cIMT [14]. To date, the association of ambient air pollution and MRI-captured CWV has not been studied. In the Canadian Alliance for Healthy Hearts and Minds Cohort (CAHHM) [15] of generally healthy adults, we sought to characterize the associations between low levels of exposure to $PM_{2.5}$, $NO_2$, and $O_3$, and MRI-measured carotid atherosclerosis as a major CVD pathway.

## Methods

### Study design and participants

The design and methods of the CAHHM prospective cohort study have been previously described [15]. Participants were recruited from January 1, 2014 to December 31, 2018 from the provinces of British Columbia, Alberta, Ontario, Quebec, and Nova Scotia in mostly urban locations [15]. Research ethics board approval was obtained from the Hamilton Integrated Research Ethics Board (HiREB # 13–255), and all participants provided written informed consent. All data were deidentified. The cohort includes 8258 adults from across Canada, of whom > 80% were participants in ongoing prospective cohort studies and assessed for CVD traditional risk factors. MRI scans of the brain, heart, carotid artery, and abdomen were performed at enrollment, and 7973 participants completed a standard carotid MRI scan. Adults with known history of CVD (defined as a self-reported history of stroke, coronary artery disease, heart failure, or other heart disease), incomplete data on the non-lab based cardiovascular risk score, or incomplete air pollutants values were excluded for the presented analyses, leaving a final sample size of 6645 participants (S1 Fig).

### Assessment of air pollution exposure

The three major air pollutants of interest in this study were $PM_{2.5}$, $NO_2$, and $O_3$. The development of these exposure datasets has been documented elsewhere [16–18] and they have been used in multiple Canadian epidemiological studies, including the recent Mortality-Air Pollution Associations in Low Exposure Environments (MAPLE) study [19]. Briefly, annual average exposures for the five years prior to the start of CAHHM recruitment (2008–2012), data distributed by the Canadian Urban Environmental Health Research Consortium (CANUE) [20] (www.canue.ca), were linked to CAHHM using the six-character residential postal code of participants at the time of recruitment. The average exposure over the five years prior to recruitment was chosen as it is considered to be representative of long-term air pollution exposure gradients and relevant for investigating subclinical CVD, which manifests over a long period of time [21].

Key emission sources for $PM_{2.5}$ are industrial emissions, wildfire smoke, space heating, residential wood heating, cooking, agriculture, and vehicle traffic emissions. A significant fraction of $PM_{2.5}$ is a result of atmospheric chemistry, forming from a range of gaseous precursors such as sulphur dioxide ($SO_2$), $NO_2$, ammonia ($NH_4$), and volatile and semi volatile organic compounds. Yearly averages of $PM_{2.5}$ concentrations prior to baseline assessment were estimated across a 1x1 kilometer grid covering North America using NASA MODIS, MISR, and Sea-WIFS satellite instruments, with aerosol vertical profiles and scattering properties simulated

by the GEOS-Chem chemical transport model [16]. To adjust for any residual bias in the satellite-derived $PM_{2.5}$ estimates, a geographically weighted regression (GWR) incorporating ground-based observations was then applied [16]. Good agreement was found with cross-validated surface observations across North-America ($R^2$ = 0.70). For most residential addresses, postal code areas were considerably smaller than 1x1 km so that the assigned $PM_{2.5}$ concentration matches the 1x1 km grid square that the postal code is found within. Specifically, assigning $PM_{2.5}$ to postal codes was performed using the single linkage approach where the $PM_{2.5}$ grid square selected was the one closest to the x, y coordinate within a postal code polygon that best represents where the majority of the population lived.

$NO_2$ is considered an indicator of TRAP, which is a complex mixture of gases and particles, including ultrafine particles (diameter $\leq$ 0.1 μm). Annual average $NO_2$ concentrations in parts per billion (ppb) were estimated for each postal code location using a national land use regression (LUR) model for the year 2006 [17] at 100 m resolution and adjusted for prior and subsequent years using long-term air quality monitoring station data. The LUR $NO_2$ model included road length, 2005–2011 satellite $NO_2$ estimates, area of industrial land use within 2 km, and summer rainfall as predictors of regional $NO_2$ variation [17]. Deterministic gradients were used to model local scale variation related to roads (i.e. traffic) [17]. The final $NO_2$ model showed good performance, explaining 73% of the variation in measurements from national air pollution surveillance (NAPS) monitoring data with a root mean square error (RMSE) of 2.9 ppb.

$O_3$ is a photochemically produced oxidant gas that results from the reaction between sunlight and NOx and VOCs emitted from various natural sources and human activities such as fossil fuel combustion and wood combustion. Annual mean concentrations of $O_3$ exposure at 10–15 km resolution were estimated using the GEM-MACH (Global Environmental Multiscale—Modelling Air Quality and Chemistry) air quality forecast model combined with observations from monitoring networks [18, 22].

## Subclinical MRI outcomes

Details of the CAHHM MRI protocol have been previously published [15]. The protocol used validated standard techniques to collect information on morphology, function and tissue characteristics. Briefly, participants underwent a short non-contrast enhanced scan using a 1.5 or 3.0 Tesla magnet. Each of the centres underwent a validated test scan for quality assurance. Carotid artery vessel wall volume ($mm^3$) (left, right, and combined) within a 32-mm vessel length centred on each carotid bifurcation (to include distal common and proximal internal carotid arteries) was measured by subtracting lumen volume from total vessel volume. The lumen was defined semi-automatically from axial bright blood images of the time of flight sequence which were reconstructed at 2 mm intervals. The outer wall of the carotid artery was semi-automatically defined and adjusted as needed by expert readers. The area of the vessel wall in each image was estimated by subtracting the lumen area from the outer wall vessel area. Vessel wall volume per slice was calculated by multiplying by 2 mm per slice. Vessel wall volumes for right and left carotid arteries were estimated by integrating the volume for the total number of slices for each artery. We used the maximum of either the left or right CWV as the measure of atherosclerosis in this study.

## The INTERHEART risk score

The non-laboratory-based INTERHEART risk score is a validated tool developed to estimate a person's myocardial infarction (MI) risk based on a compilation of risk factors [23]. These include age, sex, smoking, second-hand smoke exposure, diabetes, high blood pressure, and

family history of MI, waist-to-hip ratio (WHR); home or work social stress, depression, simple dietary questions, and physical activity [1]. The score ranges from 0 to 48 and is categorized into low- (a score $\leq$ 9), moderate- (a score of 10 to 15), or high- (a score $\geq$ 16) risk categories and is significantly associated with diagnosed CVD, and also the presence of subclinical cerebrovascular disease without known clinical CVD [11, 23].

## Definitions

Individual socioeconomic status: Education was categorized as the highest level of education attainment (High school or less, College or Trade, or University Degree). Employment status was categorized as employed, retired or unemployed. An indicator variable was used to identify individuals who traveled outside of their community of residence for work.

Neighbourhood socioeconomic status: Area-based social deprivation index and material deprivation index from 2011 Canadian census data linked to participants six-character postal code through CANUE databases were used to represent the socioeconomic status of the local community/society to which participants belonged [24]. The indices are the first two components of a principal component analysis (PCA) of the following six variables: the proportion of persons without a high school diploma; the employment population ratio; the average personal income; the proportion of persons living alone; the proportion of individuals separated, divorced or widowed; and the proportion of single-parent families [24].

Neighbourhood walkability: Walkability measures the degree to which a neighbourhood supports walking and was included because, in a nationally representative Canadian study (n = 1.8 million participants) with a 15-year follow-up, neighbourhood walkability was associated with reduced risk of cardiovascular mortality (HR: 0.91 [0.88, 0.95]) [25]. Participants residential postal codes were linked to the Canadian Active Living Environments Index (Can-ALE) categorical variable that characterizes the favourability/friendliness of active living (i.e. walkability) potential of neighbourhoods in census metropolitan areas (CMA) on a scale from 1 (very low) to 5 (very high) based on intersection density, dwelling density, and points of interest measures available for the year 2016. An environment with a very high walkability tends to be densely populated and has very connected street patterns and a variety of walking destinations. Consequently, such environments are more urbanized and tend to experience higher $NO_2$ levels. More information on Can-ALE is available at: http://canue.ca/wp-content/uploads/2018/03/CanALE_UserGuide.pdf.

## Statistical analysis

The distribution of continuous variables is presented as means with standard deviation, and categorical variables are presented as counts and percentages. We assessed the association of the continuous measure of air pollutant exposure with CWV, expressed as a 5 μg/m$^3$ increment for $PM_{2.5}$, a 5 ppb increment for $NO_2$, and a 3 ppb increment for $O_3$, as previously reported for relevant air pollutant concentrations in developed countries [8, 9]. The associations were explored using 5 linear mixed models, each with random intercepts representing the effect of recruitment centre. This random intercept was a proxy for spatial clustering of participants. We considered a parsimonious set of demographic, lifestyle, and environmental characteristics as potential covariates in the association between CWV and air pollution, considering collinearity. Our final variable selection was guided by published literature, previous knowledge, as well as variables that we observed to be effect modifiers of observed associations. Neighbourhood greenness did not meet the threshold for covariate selection and neighbourhood noise could not be tested; however, prior studies with noise adjustment showed stable if not larger effect estimates of association [9, 10]. The following fixed covariates were included

in each model: 1) none ("unadjusted model"); 2) participant's age, sex, and ethnicity ("basic model"); 3) further adjusted for contributing individual-level factors *i.e.*, the INTERHEART risk score, education, and working outside of the lived-in community ("lifestyle model"); **4) further adjusted for community-level factors i.e. walkability, material factor score, and social factor score (our a priori "primary model")** and 5) model further adjusted for co-pollutants within two-pollutant models at a time ("co-pollutant model"). A complete case analysis was employed because missing data on covariates was low (1.6% for education and 0.08% for Can-ALE index). In sensitivity analyses, models 1–5 were i. stratified by sex and ii. repeated after excluding participants based on immigration status for those who had been in Canada for less than ten years (n = 5885), and iii. stratified based on workplace location (residing at home/working in the lived-in community versus working outside the lived-in community) to address possible exposure misclassification due to major time away from residence. To investigate interactions between the pollutants, we modeled the effect of one pollutant on CWV within low, medium, and high levels of a second pollutant with the same fixed covariates of models 1–4 and tested the statistical significance of the interaction term of the two pollutants. A 2-sided p <0.05 was considered nominally significant with no adjustment for multiple testing. All analyses were completed using SAS software, version 9.4 (SAS Institute Inc).

## Results

### Participant characteristics

Of the 6645 participants enrolled in this study, 3253 (48.9%) were from Ontario, 1575 (23.7%) from Quebec, 744 (11.2%) from British Columbia, 671 (10.1%) from Nova Scotia, and 402 (6.0%) from Alberta. For all the regions, over 92% of the cohort's postal codes were in urban areas. The mean age of participants at enrolment was 57.6 years (SD = 8.8; range = 32–81 years) and 56.0% of participants were women. The mean CWV of participants was 900.1 $mm^3$ (165.1) at the time of the MRI scan. Demographic, anthropometric, and lifestyle characteristics of the study participants are found in Tables 1 and 2. The mean (SD) 5-year pollutant concentrations immediately preceding enrolment for $PM_{2.5}$ was 6.9 $\mu g/m^3$ (2.0), ranging from the lowest [3.2 $\mu g/m^3$ (0.5)] in parts of the Calgary, Alberta region to the highest [8.6 $\mu g/m^3$ (1.5)] in London, Ontario; for $NO_2$ was 12.9 ppb (5.9), ranging from lowest [4.1 ppb (1.2)] in Halifax, Nova Scotia to highest [17.0 ppb (3.9)] in Toronto, Ontario; and for $O_3$ was 24.6 ppb (4.0), ranging from lowest [16.9 ppb (3.0)] in Vancouver, British Columbia to highest [30.4 ppb (1.2)] in London, Ontario, as presented in Table 3. Participant characteristics stratified by sex are presented in S1–S3 Tables.

### Long-term pollutant exposure and MRI-measured CWV–Primary analysis

Associations of 5-year pollutant exposures with subclinical atherosclerosis as measured by CWV are presented in Fig 1 and Table 4. The association between $PM_{2.5}$ and CWV was inconsistent and not statistically significant in the primary model. A 5 $\mu g/m^3$ higher $PM_{2.5}$ concentration was not associated with CWV (mean = -8.4 $mm^3$; 95% CI -23.2, 6.48; p = 0.27). A 5 ppb higher $NO_2$ concentration was associated with 11.8 $mm^3$ lower CWV (95% CI -16.2, -7.31; p<0.0001); contradictory to our hypothesis. A 3 ppb increase in $O_3$ was associated with 9.34 $mm^3$ higher CWV (95% CI 4.75, 13.92; p<0.0001). These results remained consistent in the sensitivity analysis in males and females, after excluding immigrants with less than 10 years of residence in Canada, and after excluding participants working away from their lived-in community (Fig 1, Tables 4 and 5). Of note, Pearson correlation between $PM_{2.5}$ and $NO_2$ was [r = +0·39; p<·0001], between $PM_{2.5}$ and $O_3$ was [r = +0·16; p<·0001], and between $NO_2$ and $O_3$ was [r = -0·23; p<·0001].

**Table 1. Anthropometric characteristics of the study population by region of Canada.**

| | N | Overall | Region of Canada | | | | |
|---|---|---|---|---|---|---|---|
| | | | BC | AB | ON | QC | NS |
| Number of participants | 6645 | 6645 | 744 | 402 | 3253 | 1575 | 671 |
| Women, n (%) | 6645 | 3718 (56.0) | 408 (54.8) | 197 (49.0) | 1948 (59.9) | 811 (51.5) | 354 (52.8) |
| Age, y | 6645 | 57.6 (8.8) | 56.8 (8.8) | 53.3 (8.8) | 57.5 (9.0) | 58.6 (7.8) | 59.0 (9.3) |
| Weight, kg | 6645 | 76.3 (16.5) | 72.5 (15.9) | 80.7 (17.5) | 75.7 (16.5) | 77.4 (16.4) | 78.1 (15.4) |
| Height, cm | 6645 | 168.5 (9.4) | 167.5 (9.6) | 173.2 (10.1) | 168.3 (9.1) | 167.6 (9.3) | 169.4 (9.2) |
| **Body Mass Index, mean (SD), kg/m$^2$** | 6645 | 26.8 (4.9) | 25.7 (4.4) | 26.8 (4.9) | 26.6 (4.9) | 27.5 (5.0) | 27.2 (4.8) |
| <25 (Normal), n (%) | 6645 | 2657 (40.0) | 352 (47.3) | 163 (40.5) | 1366 (42.0) | 535 (34.0) | 241 (35.9) |
| 25–29 (Overweight), n (%) | 6645 | 2536 (38.2) | 285 (38.3) | 150 (37.3) | 1193 (36.7) | 647 (41.1) | 261 (38.9) |
| 30+ (Obese), n (%) | 6645 | 1452 (21.9) | 107 (14.4) | 89 (22.1) | 694 (21.3) | 393 (25.0) | 169 (25.2) |
| Percent Body Fat, % | 6624 | 30.6 (9.2) | 28.9 (8.4) | 29.9 (9.1) | 30.9 (9.4) | 31.3 (8.8) | 30.0 (9.6) |
| Waist, cm | 6645 | 88.6 (13.7) | 84.3 (12.8) | 91.0 (13.9) | 87.5 (13.6) | 90.3 (14.0) | 93.0 (12.5) |
| Hip, cm | 6645 | 101.3 (10.7) | 99.2 (9.2) | 103.6 (10.3) | 100.4 (11.5) | 102.3 (9.7) | 104.7 (9.2) |
| Waist to hip ratio | 6645 | 0.87 (0.08) | 0.85 (0.09) | 0.88 (0.08) | 0.87 (0.08) | 0.88 (0.09) | 0.89 (0.08) |
| Waist circumference obese, n (%) | 6645 | 1924 (29.0) | 118 (15.9) | 130 (32.3) | 917 (28.2) | 492 (31.2) | 267 (39.8) |
| **Blood Pressure** | | | | | | | |
| Systolic, mmHg | 6645 | 129 (17) | 126 (17) | 127 (14) | 128 (17) | 131 (16) | 133 (16) |
| Diastolic, mmHg | 6645 | 80 (10) | 80 (10) | 85 (9) | 78 (10) | 80 (9) | 80 (10) |
| Heart rate, beats/minute | 6644 | 70.4 (11.0) | 69.4 (11.0) | 70.5 (11.0) | 71.1 (11.1) | 69.7 (10.7) | 69.6 (10.6) |
| **MRI-measured Outcomes** | | | | | | | |
| Carotid wall volume, mm$^3$ | 6645 | 900.1 (165.1) | 897.9 (170.5) | 889.9 (170.5) | 906.0 (163.2) | 893.2 (155.8) | 896.2 (184.3) |

Presented data are means (SD) unless otherwise indicated.

## Effect of co-pollutant interactions on CWV

Across the primary models summarized in Fig 2 for testing the effect of one pollutant on CWV within low, medium, and high levels of a second pollutant, there was no evidence of interaction between $PM_{2.5}$ and $O_3$ or between $PM_{2.5}$ and $NO_2$. However, the association of $NO_2$ with CWV differed according to the exposure levels of $O_3$ (p<0.0001 for interaction) and the association of $O_3$ with CWV differed according to the $NO_2$ levels (p<0.0001 for interaction). The positive association of $O_3$ with CWV within low and medium levels of $NO_2$ was not observed within high levels of $NO_2$ and the inverse association of $NO_2$ with CWV within medium and high levels of $O_3$ was not observed within low levels of $O_3$. Thus, the gaseous pollutants emerged as mutual effect modifiers (Tables 6–8).

## Discussion

In this cohort study among 6645 Canadian healthy adults using spatially resolved pollutant concentrations and comprehensive covariate adjustments, long-term exposure to air pollution was inconsistently associated with CWV as measured by MRI. $PM_{2.5}$ associations with CWV were inconsistent while $NO_2$ was associated with decreased CWV; a finding that was contradictory to our expectation. Exposure to ground-level $O_3$ was associated with increased CWV (noting the limitation that $O_3$ resolution is 10 km). These findings were consistent between men and women and remained robust even after exclusion of recent immigrants. Lastly, $NO_2$ and $O_3$ mutually modified the associations of these pollutants with CWV.

**Table 2. Demographics & lifestyle characteristics of the study population by region of Canada.**

| | | | Region of Canada | | | | |
|---|---|---|---|---|---|---|---|
| | N | Overall | BC | AB | ON | QC | NS |
| Number of participants | 6645 | 6645 | 744 | 402 | 3253 | 1575 | 671 |
| Women | 6645 | 3718 (56.0) | 408 (54.8) | 197 (49.0) | 1948 (59.9) | 811 (51.5) | 354 (52.8) |
| Age, mean (SD), y | 6645 | 57.6 (8.8) | 56.8 (8.8) | 53.3 (8.8) | 57.5 (9.0) | 58.6 (7.8) | 59.0 (9.3) |
| **Self-reported ethnicity** | | | | | | | |
| East & South East Asian | 6645 | 894 (13.5) | 282 (37.9) | 14 (3.5) | 585 (18.0) | 8 (0.5) | 5 (0.7) |
| South Asian | 6645 | 223 (3.4) | 67 (9.0) | 3 (0.7) | 148 (4.5) | 1 (0.1) | 4 (0.6) |
| White | 6645 | 5387 (81.1) | 364 (48.9) | 381 (94.8) | 2449 (75.3) | 1545 (98.1) | 648 (96.6) |
| Other[a] | 6645 | 141 (2.1) | 31 (4.2) | 4 (1.0) | 71 (2.2) | 21 (1.3) | 14 (2.1) |
| **Highest Education Attained** | | | | | | | |
| High school or less | 6541 | 842 (12.9) | 100 (13.5) | 44 (10.9) | 344 (10.6) | 290 (18.4) | 64 (10.9) |
| College or Trade | 6541 | 2107 (32.2) | 255 (34.5) | 117 (29.1) | 891 (27.5) | 671 (42.6) | 173 (29.4) |
| University Degree | 6541 | 3592 (54.9) | 385 (52.0) | 241 (60.0) | 2001 (61.8) | 613 (38.9) | 352 (59.8) |
| **Smoke status** | | | | | | | |
| Current | 6645 | 352 (5.3) | 29 (3.9) | 18 (4.5) | 156 (4.8) | 119 (7.6) | 30 (4.5) |
| Former | 6645 | 2241 (33.7) | 178 (23.9) | 132 (32.8) | 971 (29.8) | 732 (46.5) | 228 (34.0) |
| Never | 6645 | 4052 (61.0) | 537 (72.2) | 252 (62.7) | 2126 (65.4) | 724 (46.0) | 413 (61.5) |
| **Living with partner/married** | 6537 | 4938 (75.5) | 567 (76.6) | 314 (78.1) | 2431 (75.2) | 1141 (72.4) | 485 (82.5) |
| **Employment** | | | | | | | |
| Full or part time | 6534 | 4624 (70.8) | 554 (74.7) | 324 (80.6) | 2208 (68.2) | 1124 (71.5) | 414 (71.4) |
| Retired | 6534 | 1436 (22.0) | 143 (19.3) | 41 (10.2) | 727 (22.5) | 385 (24.5) | 140 (24.1) |
| No paid work | 6534 | 474 (7.3) | 45 (6.1) | 37 (9.2) | 302 (9.3) | 64 (4.1) | 26 (4.5) |
| **Individual Social disadvantage score** | | | | | | | |
| Social disadvantage Index, mean (SD) | 6121 | 1.2 (1.3) | 1.2 (1.3) | 0.8 (1.1) | 1.2 (1.3) | 1.4 (1.3) | 1.1 (1.3) |
| Low disadvantage | 6121 | 3677 (60.1) | 435 (63.0) | 283 (73.7) | 1758 (59.1) | 858 (56.9) | 343 (61.0) |
| Moderate disadvantage | 6121 | 2072 (33.9) | 209 (30.3) | 92 (24.0) | 1031 (34.6) | 542 (35.9) | 198 (35.2) |
| High disadvantage | 6121 | 372 (6.1) | 46 (6.7) | 9 (2.3) | 187 (6.3) | 109 (7.2) | 21 (3.7) |
| **INTERHEART risk score** mean (SD) | 6645 | 10.0 (5.7) | 9.2 (5.5) | 9.6 (5.7) | 10.1 (5.7) | 10.3 (5.8) | 10.2 (5.6) |
| Low | 6645 | 3392 (51.0) | 418 (56.2) | 217 (54.0) | 1645 (50.6) | 785 (49.8) | 327 (48.7) |
| Moderate | 6645 | 2123 (31.9) | 223 (30.0) | 121 (30.1) | 1069 (32.9) | 491 (31.2) | 219 (32.6) |
| High | 6645 | 1130 (17.0) | 103 (13.8) | 64 (15.9) | 539 (16.6) | 299 (19.0) | 125 (18.6) |
| **Usual workplace location** | | | | | | | |
| Outside home community | 6605 | 2301 (34.8) | 269 (36.5) | 197 (49.0) | 1008 (31.1) | 675 (43.3) | 152 (22.8) |

Presented data are n (%) unless otherwise specified.

[a] Includes Blacks, Indigenous, Mixed and unknown ethnicity

There is limited evidence that direct biological measures of early CVD are influenced by exposure to air pollution at low levels. However, the MAPLE study and MAPLE phase 2 report associations between nonaccidental mortality and cardiovascular-related mortality and long-term exposure to ambient $PM_{2.5}$ levels, including at concentrations below national air quality standards [19, 26]. The evidence is adequate for overall TRAP association with CVD mortality, but remains inconclusive for CVD morbidity *i.e.*, the effect of air pollution on traditional cardiovascular risk factors [9, 27]. The 2022 HEI report concludes that additional studies are needed for cardiometabolic outcomes and subclinical biomarkers [10].

Ultrasound measurement of cIMT is widely used because it is inexpensive, non-invasive, and does not involve exposure to a magnetic field, but it has limited clinical usefulness beyond

**Table 3. Environmental characteristics of the study population by region of Canada.**

| | N | Overall | Region of Canada | | | | |
|---|---|---|---|---|---|---|---|
| | | | BC | AB | ON | QC | NS |
| Number of participants | 6645 | 6645 | 744 | 402 | 3253 | 1575 | 671 |
| Urban postal code | 6645 | 6390 (96.2) | 686 (92.2) | 389 (96.8) | 3179 (97.7) | 1468 (93.2) | 668 (99.6) |
| **Neighbourhood Deprivation** | | | | | | | |
| Material factor score | 6447 | -0.017 (0.041) | -0.013 (0.039) | -0.040 (0.035) | -0.016 (0.043) | -0.008 (0.039) | -0.027 (0.032) |
| Social factor score | 6447 | 0.001 (0.041) | -0.006 (0.036) | -0.008 (0.044) | -0.005 (0.042) | 0.017 (0.036) | 0.008 (0.038) |
| **Walkability Measures** | | | | | | | |
| ALE Index | 6640 | 1.806 (4.501) | 1.535 (3.661) | 0.810 (1.854) | 2.618 (5.610) | 1.172 (2.976) | 0.255 (1.764) |
| ALE Index Class, n (%) | 6640 | | | | | | |
| Class 1: very low | 6640 | 772 (11.6) | 81 (10.9) | 40 (10.0) | 300 (9.2) | 232 (14.7) | 119 (17.7) |
| Class 2 | 6640 | 1917 (28.9) | 176 (23.7) | 193 (48.0) | 820 (25.2) | 472 (30.0) | 256 (38.2) |
| Class 3 | 6640 | 2014 (30.3) | 249 (33.6) | 146 (36.3) | 1079 (33.2) | 326 (20.7) | 214 (31.9) |
| Class 4 | 6640 | 1182 (17.8) | 166 (22.4) | 12 (3.0) | 611 (18.8) | 311 (19.8) | 82 (12.2) |
| Class 5: very high | 6640 | 755 (11.4) | 70 (9.4) | 11 (2.7) | 442 (13.6) | 232 (14.7) | 0 (0.0) |
| **Linked Air Quality Measures** | | | | | | | |
| $PM_{2.5}$, ug/m$^3$, over 2008–2012 | 6645 | 6.9 (2.0) | 6.7 (1.4) | 3.2 (0.5) | 8.2 (1.5) | 6.0 (1.5) | 5.2 (1.1) |
| $NO_2$, ppb, over 2008–2012 | 6645 | 12.9 (5.9) | 15.2 (4.8) | 12.8 (3.5) | 14.1 (5.3) | 13.0 (5.8) | 4.1 (1.2) |
| $O_3$, ppb, over 2008–2012 | 6645 | 24.6 (4.0) | 17.8 (4.3) | 23.4 (2.1) | 27.1 (2.6) | 23.9 (2.4) | 22.3 (1.3) |

Presented data are means (SD) unless otherwise indicated. CMA: census metropolitan area; ALE: active living environment

traditional CVD risk factors [20]. Carotid wall thickness, determined by MRI, generates more reproducible measurements than ultrasound and is a superior measure of early plaque development [12, 13]—ideal for the investigation of healthy middle-aged populations. It is also more consistently associated with incident CVD than ultrasound-measured cIMT [14]. Compared to cIMT, CWV includes the adventitia, the source of vasa vasorum [12]. In terms of sensitivity of MRI-measured CWV, studies have suggested adventitial thickening to be an early sign of atherosclerosis, whereas a dense network of adventitial vasa vasorum can signify progression of atherosclerosis to symptomatic disease [12]. We have previously shown in CAHHM that simple cardiovascular risk scores were significantly associated with CWV, where mean (SD) CWV for low, medium, and high INTERHEART risk score categories were 881.5 (163.1), 915.4 (166.6), and 940.9 (172.9) mm$^3$, respectively [11]. Therefore, the overall mean in the current analysis set (900 mm$^3$) corresponds to someone with a low-moderate INTERHEART risk score. The inconsistent association between $PM_{2.5}$ and a state-of-the-art surrogate marker for subclinical atherosclerosis we observed across Canada parallels previous findings of the Multicultural Community Health Assessment Trial (M-CHAT) study between multiple particle and gaseous measures of TRAP (e.g., $NO_2$, $PM_{2.5}$, black carbon) and progression of carotid artery atherosclerosis in Vancouver, BC.[6] This suggests that $PM_{2.5}$ is not affecting CV risk through early atherosclerosis formation. Therefore, if there is an increased risk with exposure to $PM_{2.5}$, it may operate through other intermediary pathways such as release of proinflammatory mediators, autonomic nervous system perturbations, and translocation of particle constituents into the blood, acting independently from the promotion of plaque build-up [28]. We therefore recommend employing CWV over cIMT, especially in relatively low exposure settings such as Canada where a more precise measurement of subclinical atherosclerosis may be of higher yield. Of note, walkability was shown to modulate the effects of $PM_{2.5}$ in the main and secondary analyses; a neighbourhood characteristic that has been scarcely captured in previous literature.

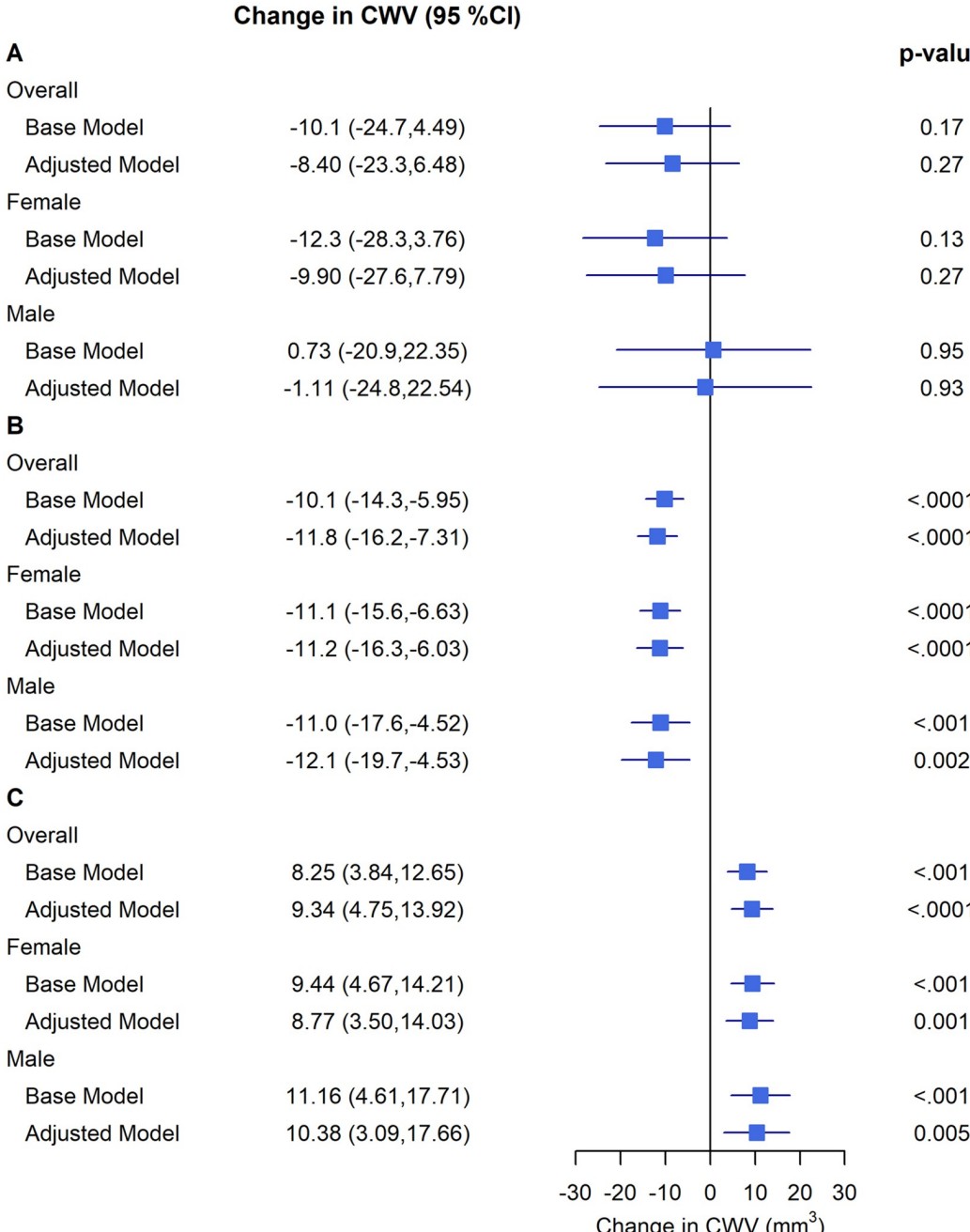

**Fig 1. Associations of 5-year pollutant exposures with subclinical atherosclerosis as measured by carotid wall volume (CWV).** Presented data are carotid wall volume adjusted estimates (95% CI) per (A) 5 µg/m$^3$ increase for PM$_{2.5}$, (B) 5 ppb increase for NO$_2$, and (C) 3 ppb increase for O$_3$, from a linear mixed model with centre modelled as a random intercept. The base model has no fixed effects; the adjusted model includes age, (sex), ethnicity, INTERHEART risk score, education, workplace location, walkability categories, material factor score, and social factor score.

The weak inverse association of NO$_2$ (an indicator of TRAP and/or combustion pollution in general) with CWV in our study is unexpected and needs careful evaluation. There is insufficient collective evidence in the literature linking NO$_2$ or TRAP with subclinical atherosclerosis in healthy adults specifically. The evidence of an association between TRAP and

**Table 4. Association of carotid wall volume (mm$^3$) with pollutants exposure.**

| | Overall (n = 6223) | | Females (n = 3472) | | Males (n = 2751) | | Remove recent immigrants (n = 5885) | |
|---|---|---|---|---|---|---|---|---|
| | Effect (95% CI) | p- | Effect (95% CI) | p- | Effect (95% CI) | p- | Effect (95% CI) | p- |
| **PM$_{2.5}$** | | | | | | | | |
| Model 1 | -10.1 (-24.7,4.49) | 0.1747 | -12.3 (-28.3,3.76) | 0.1333 | 0.73 (-20.9,22.35) | 0.9469 | -12.5 (-27.6,2.71) | 0.1075 |
| Model 2 | -14.9 (-28.4,-1.42) | 0.0303 | -17.9 (-33.9,-1.87) | 0.0286 | -6.46 (-28.2,15.32) | 0.5610 | -18.2 (-32.1,-4.21) | 0.0107 |
| Model 3 | -15.0 (-28.5,-1.55) | 0.0288 | -18.5 (-34.5,-2.46) | 0.0238 | -6.18 (-27.9,15.58) | 0.5777 | -18.4 (-32.3,-4.41) | 0.0099 |
| Model 4 | -8.40 (-23.3,6.48) | 0.2685 | -9.90 (-27.6,7.79) | 0.2728 | -1.11 (-24.8,22.54) | 0.9269 | -12.5 (-27.9,2.96) | 0.1130 |
| Model 5:+NO$_2$ | 2.26 (-13.2,17.70) | 0.7747 | 0.94 (-17.5,19.37) | 0.9207 | 8.84 (-15.6,33.23) | 0.4775 | -2.32 (-18.3,13.66) | 0.7762 |
| Model 5:+O$_3$ | 0.60 (-14.7,15.91) | 0.9391 | -1.35 (-19.2,16.46) | 0.8818 | 6.28 (-17.9,30.45) | 0.6103 | -2.49 (-18.3,13.31) | 0.7577 |
| **NO$_2$** | | | | | | | | |
| Model 1 | -10.1 (-14.3,-5.95) | <.0001 | -11.1 (-15.6,-6.63) | <.0001 | -11.0 (-17.6,-4.52) | 0.0009 | -10.2 (-14.5,-5.89) | <.0001 |
| Model 2 | -11.2 (-15.0,-7.32) | <.0001 | -11.1 (-15.6,-6.66) | <.0001 | -10.7 (-17.2,-4.14) | 0.0014 | -11.2 (-15.2,-7.25) | <.0001 |
| Model 3 | -11.0 (-14.9,-7.10) | <.0001 | -11.6 (-16.1,-7.08) | <.0001 | -9.96 (-16.5,-3.38) | 0.0030 | -11.0 (-15.0,-7.02) | <.0001 |
| Model 4 | -11.8 (-16.2,-7.31) | <.0001 | -11.2 (-16.3,-6.03) | <.0001 | -12.1 (-19.7,-4.53) | 0.0018 | -11.9 (-16.5,-7.37) | <.0001 |
| Model 5:+O$_3$ | -9.66 (-14.8,-4.55) | 0.0002 | -9.42 (-15.4,-3.47) | 0.0019 | -9.13 (-17.5,-0.76) | 0.0326 | -8.85 (-14.1,-3.60) | 0.0010 |
| Model 5:+PM$_{2.5}$ | -11.9 (-16.5,-7.32) | <.0001 | -11.2 (-16.6,-5.88) | <.0001 | -12.8 (-20.6,-4.96) | 0.0014 | -11.8 (-16.5,-7.04) | <.0001 |
| **O$_3$** | | | | | | | | |
| Model 1 | 8.25 (3.84,12.65) | 0.0002 | 9.44 (4.67,14.21) | 0.0001 | 11.16 (4.61,17.71) | 0.0008 | 9.26 (4.76,13.76) | <.0001 |
| Model 2 | 10.45 (6.34,14.55) | <.0001 | 9.86 (5.13,14.59) | <.0001 | 11.01 (4.41,17.60) | 0.0011 | 11.59 (7.41,15.77) | <.0001 |
| Model 3 | 10.36 (6.23,14.48) | <.0001 | 10.27 (5.52,15.02) | <.0001 | 10.44 (3.82,17.05) | 0.0020 | 11.49 (7.29,15.69) | <.0001 |
| Model 4 | 9.34 (4.75,13.92) | <.0001 | 8.77 (3.50,14.03) | 0.0011 | 10.38 (3.09,17.66) | 0.0053 | 10.82 (6.14,15.50) | <.0001 |
| Model 5:+NO$_2$ | 4.27 (-1.06,9.60) | 0.1167 | 3.47 (-2.76,9.69) | 0.2746 | 6.66 (-1.39,14.70) | 0.1047 | 6.25 (0.82,11.68) | 0.0241 |
| Model 5:+PM$_{2.5}$ | 9.39 (4.62,14.15) | 0.0001 | 8.64 (3.22,14.06) | 0.0018 | 10.81 (3.34,18.28) | 0.0046 | 10.62 (5.77,15.47) | <.0001 |

Presented data are adjusted estimates (95% CI) per 5 μg/m$^3$ increase for PM$_{2.5}$, 5 ppb increase for NO$_2$, and 3 ppb increase for O$_3$, from a linear mixed model with centre modelled as a random intercept. Model 1, unadjusted. Model 2, adjusted for age, (sex), and ethnicity. Model 3 further adjusted for the INTERHEART risk score and education and workplace location. Primary model 4 further adjusted for walkability categories, material factor score and social factor score. Model 5 further adjusted for co-pollutants.

cardiovascular morbidity is low [9]. Three of 5 studies (n = 144,787) included in a meta-analysis of NO$_2$ and ischemic heart disease (IHD) [10] incidence showed a positive association (a prospective cohort conducted across 11 European cohorts [the European Study of Cohorts for Air Pollution Effects: ESCAPE] [29]; Athens, Greece [30]; and Oakland, California [31]), while two studies (n = 663,751) reported a negative association (London, UK [32]; Vancouver, BC [6]). In fact, the meta-analytical summary estimate, relative risk of IHD events per 10 μg/m$^3$ NO$_2$, was 0.99 (0.94;1.05). MESA-Air did not find a relationship between NO$_2$ or other pollutant exposures and cIMT change, instead exposure was positively associated with coronary artery calcification progression [2]. In four European cohort, ESCAPE findings were inconsistent for an association between NO$_2$ and cIMT, in fact, all four cohorts and their meta-analytical estimate, showed an inverse association, similar to our observation in CAHHM [33]. However, in a study of a *high cardiovascular risk* population (2227 patients mean age of 62.9 years) in London, Ontario, NO$_2$ exposure was associated with cumulative plaque burden as captured by carotid total plaque area (TPA) using two-dimensional ultrasound [34]. Collectively within the existing body of literature, it is plausible that NO$_2$ is probably not involved in early carotid thickening but perhaps in more advanced morbid stages.

Our finding on the effect of chronic ambient O$_3$ exposure on subclinical atherosclerosis is congruent with what the MESA study had previously reported on progression of IMT of the

**Table 5. Association of carotid wall volume (mm³) with pollutants exposure stratified by workplace location.**

|  | Overall (n = 6223) | | Working away (n = 2188) | | Work in community (n = 4035) | |
|---|---|---|---|---|---|---|
|  | Effect (95% CI) | p- | Effect (95% CI) | p- | Effect (95% CI) | p- |
| **PM₂.₅** |  |  |  |  |  |  |
| Model 1 | -10.1 (-24.7,4.49) | 0.1747 | -1.27 (-24.7,22.21) | 0.9157 | -13.2 (-31.1,4.67) | 0.1476 |
| Model 2 | -14.9 (-28.4,-1.42) | 0.0303 | -5.83 (-27.8,16.16) | 0.6034 | -16.3 (-32.9,0.25) | 0.0536 |
| Model 3 | -15.0 (-28.5,-1.55) | 0.0288 | -5.50 (-27.5,16.44) | 0.6229 | -16.8 (-33.4,-0.20) | 0.0473 |
| Model 4 | -8.40 (-23.3,6.48) | 0.2685 | 0.87 (-22.8,24.55) | 0.9429 | -9.81 (-28.3,8.64) | 0.2974 |
| Model 5:+NO₂ | 2.26 (-13.2,17.70) | 0.7747 | 3.77 (-20.8,28.29) | 0.7630 | 5.21 (-14.0,24.40) | 0.5944 |
| Model 5:+O₃ | 0.60 (-14.7,15.91) | 0.9391 | 5.53 (-18.2,29.26) | 0.6477 | 0.67 (-18.1,19.40) | 0.9441 |
| **NO₂** |  |  |  |  |  |  |
| Model 1 | -10.1 (-14.3,-5.95) | <.0001 | -3.62 (-10.6,3.31) | 0.3059 | -13.1 (-18.2,-7.93) | <.0001 |
| Model 2 | -11.2 (-15.0,-7.32) | <.0001 | -4.35 (-10.8,2.06) | 0.1833 | -14.4 (-19.1,-9.64) | <.0001 |
| Model 3 | -11.0 (-14.9,-7.10) | <.0001 | -3.62 (-10.1,2.82) | 0.2702 | -14.5 (-19.3,-9.69) | <.0001 |
| Model 4 | -11.8 (-16.2,-7.31) | <.0001 | -3.40 (-10.9,4.15) | 0.3771 | -15.5 (-21.0,-10.1) | <.0001 |
| Model 5:+O₃ | -9.66 (-14.8,-4.55) | 0.0002 | -1.64 (-9.81,6.52) | 0.6927 | -12.9 (-19.2,-6.53) | <.0001 |
| Model 5:+PM₂.₅ | -11.9 (-16.5,-7.32) | <.0001 | -3.70 (-11.5,4.10) | 0.3524 | -16.0 (-21.6,-10.3) | <.0001 |
| **O₃** |  |  |  |  |  |  |
| Model 1 | 8.25 (3.84,12.65) | 0.0002 | 5.98 (-1.20,13.17) | 0.1023 | 9.52 (4.29,14.75) | 0.0004 |
| Model 2 | 10.45 (6.34,14.55) | <.0001 | 6.65 (-0.07,13.37) | 0.0524 | 12.58 (7.69,17.46) | <.0001 |
| Model 3 | 10.36 (6.23,14.48) | <.0001 | 6.27 (-0.45,12.98) | 0.0675 | 12.64 (7.72,17.56) | <.0001 |
| Model 4 | 9.34 (4.75,13.92) | <.0001 | 4.93 (-2.42,12.27) | 0.1884 | 11.92 (6.42,17.43) | <.0001 |
| Model 5:+NO₂ | 4.27 (-1.06,9.60) | 0.1167 | 4.14 (-3.93,12.20) | 0.3146 | 5.16 (-1.33,11.65) | 0.1191 |
| Model 5:+PM₂.₅ | 9.39 (4.62,14.15) | 0.0001 | 5.34 (-2.15,12.83) | 0.1622 | 11.97 (6.31,17.63) | <.0001 |

Presented data are adjusted estimates (95% CI) per 5 µg/m³ increase for PM₂.₅, 5 ppb increase for NO₂, and 3 ppb increase for O₃, from a linear mixed model with centre modelled as a random intercept. Model 1, unadjusted. Model 2, adjusted for age, sex, and ethnicity. Model 3 further adjusted for the INTERHEART risk score and education. Primary model 4 further adjusted for walkability categories, material factor score and social factor score. Model 5 further adjusted for co-pollutants.

common carotid artery and new carotid plaque formation with outdoor O₃ exposure in six U.S. city regions [8]. One underlying biochemical pathway might be through the formation of reactive oxygen species that further give rise to increased oxidative stress and persistent chronic systemic inflammation [8]. Yet, given the inverse association between O₃ and NO₂ generally in urban settings, this "effect" of O₃ may simply reflect the inverse of the NO₂ effect, and because the O₃ spatial resolution is 10 km, more caution is needed. The Multicenter Ozone Study in oldEr Subjects (MOSES) found that controlled exposure to low-levels of O₃ did not affect selected blood biomarkers of systemic inflammation and prothrombotic state (C-reactive protein, monocyte-platelet conjugates, and microparticle-associated tissue factor activity) [35].

Next, given the observed interaction between O₃ and NO₂, our study emphasizes the need for further investigation of different exposures in combination. It also emphasizes the value of adjusting for novel neighbourhood characteristics such as active living environment, to examine effect modification and help further investigate regional variability. Neighbourhood factors that might modify the association between air pollution and CWV include poverty/affluence, overcrowding, living in apartment buildings, commuting, and proximity to roads.

## Strengths and limitations

Our study has obtained unique health measurements of subclinical cardiovascular markers using MRI on nearly 6600 Canadians along with individual-level information on

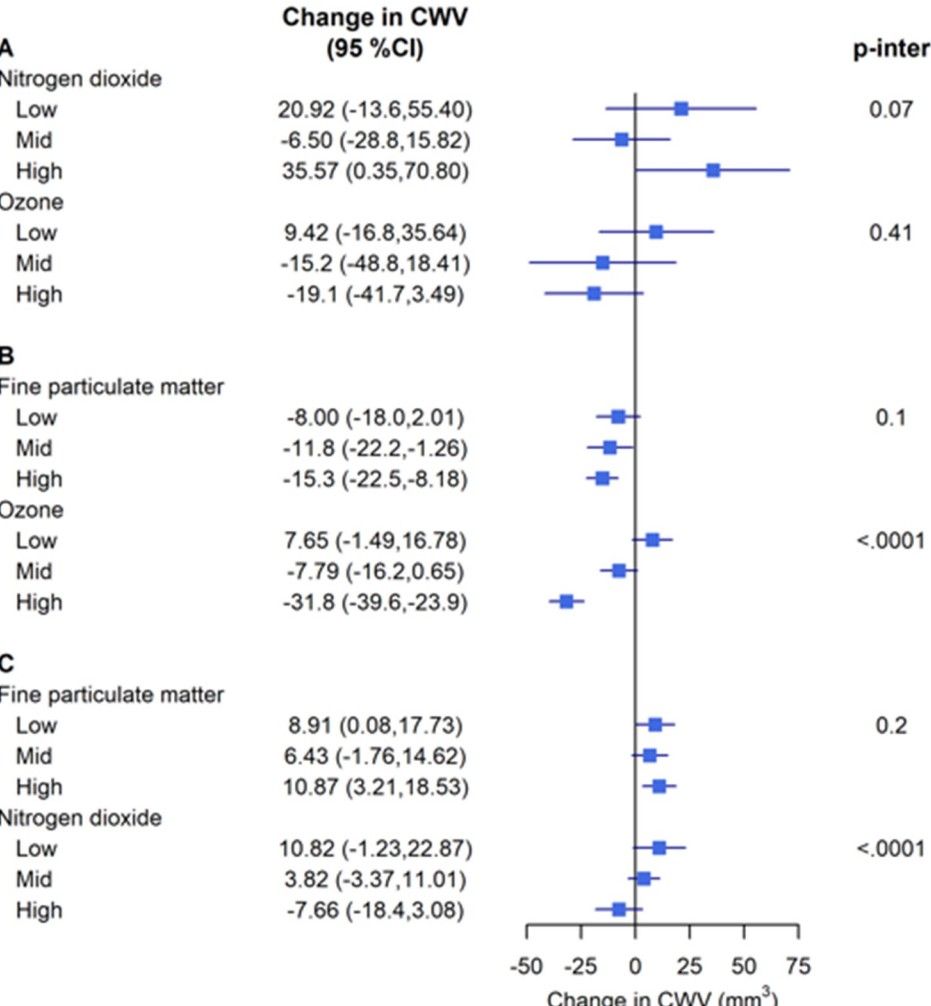

**Fig 2. Effect of copollutant interactions on carotid wall volume (CWV).** Presented data are carotid wall volume adjusted estimates (95% CI) per (A) 5 μg/m$^3$ increase for PM$_{2.5}$, (B) 5 ppb increase for NO$_2$, and (C) 3 ppb increase for O$_3$, from a linear mixed model with centre modelled as a random intercept and the following fixed effects: age, sex, ethnicity, INTERHEART risk score, education, workplace location, walkability categories, material factor score, and social factor score. Each pollutant is represented within co-pollutant tertiles and a p-value of the interaction term of the two pollutants (p-inter).

environmental factors and lifestyle known to influence cardiometabolic outcomes. While IMT has been criticized as an accurate marker of atherosclerosis with sensitivity limitations of the ultrasound methodology [34], our study uses MRI-characterized CWV to assess atherosclerosis. Moreover, compared to MESA which reported on cIMT progression in 3392 participants with low-exposure levels, our study is well-powered. Next, the cohort's diverse geographic coverage across Canada offers an exposure gradient in ambient PM$_{2.5}$, NO$_2$, and O$_3$ that parallels what has previously been explored by the national Canadian Census Health and Environment Cohort (CanCHEC) [19]. Finally, exposure to air pollution values represented a time frame prior to knowledge of the outcome for each participant, i.e., the air pollution data collected for the 5-year period prior to the MRI. Several limitations are important to mention. First, individual-level exposures were estimated based on residence address. While this is common in

**Table 6. Association of carotid wall volume (mm$^3$) with O$_3$ within co-pollutant exposure tertiles.**

| | Overall N = 6223 | | Low PM$_{2.5}$ (1.78–5.64 µg/m$^3$) N = 1903 | | Mid PM$_{2.5}$ 5.65–8.18 µg/m$^3$ N = 1898 | | High PM$_{2.5}$ 8.18–11.2 µg/m$^3$ N = 2422 | |
|---|---|---|---|---|---|---|---|---|
| | Effect (95% CI) | p- | Effect (95% CI) | p- | Effect (95% CI) | p- | Effect (95% CI) | p- |
| Model 1 | 8.25 (3.84,12.65) | 0.0002 | 7.16 (-1.00,15.32) | 0.0855 | 9.56 (2.26,16.85) | 0.0103 | 8.31 (0.24,16.39) | 0.0436 |
| Model 2 | 10.45 (6.34,14.55) | <.0001 | 9.73 (2.20,17.26) | 0.0113 | 7.52 (-0.35,15.40) | 0.0612 | 12.99 (5.55,20.44) | 0.0006 |
| Model 3 | 10.36 (6.23,14.48) | <.0001 | 9.96 (2.39,17.54) | 0.0100 | 7.51 (-0.39,15.41) | 0.0625 | 11.95 (4.49,19.42) | 0.0017 |
| Model 4 | 9.34 (4.75,13.92) | <.0001 | 8.91 (0.08,17.73) | 0.0480 | 6.43 (-1.76,14.62) | 0.1240 | 10.87 (3.21,18.53) | 0.0055 |
| | | | Low NO$_2$ (0.88–8.10 ppb) N = 1539 | | Mid NO$_2$ (8.12–15.5 ppb) N = 2202 | | High NO$_2$ (15.5–38.7 ppb) N = 2482 | |
| | | | Effect (95% CI) | p- | Effect (95% CI) | p- | Effect (95% CI) | p- |
| Model 1 | 8.25 (3.84,12.65) | 0.0002 | 12.32 (0.91,23.74) | 0.0343 | 0.56 (-7.10,8.21) | 0.8869 | -5.30 (-16.4,5.78) | 0.3483 |
| Model 2 | 10.45 (6.34,14.55) | <.0001 | 11.99 (0.90,23.08) | 0.0341 | 4.21 (-3.01,11.42) | 0.2533 | -6.42 (-17.1,4.25) | 0.2382 |
| Model 3 | 10.36 (6.23,14.48) | <.0001 | 12.63 (1.55,23.70) | 0.0255 | 4.28 (-2.93,11.48) | 0.2443 | -6.81 (-17.5,3.86) | 0.2108 |
| Model 4 | 9.34 (4.75,13.92) | <.0001 | 10.82 (-1.23,22.87) | 0.0784 | 3.82 (-3.37,11.01) | 0.2974 | -7.66 (-18.4,3.08) | 0.1622 |

Presented data are adjusted CWV estimates (95% CI) per, 3 ppb increase for O$_3$, from a linear mixed model with centre modelled as a random intercept. Model 1, unadjusted. Model 2, adjusted for age, (sex), and ethnicity. Model 3 further adjusted for the INTERHEART risk score, education and workplace location. Primary model 4 further adjusted for walkability categories, material factor score and social factor score.

epidemiological air pollution studies, exposure misclassification is inevitable because in this study participants' time away from residence and residential history were not taken into account in estimating long-term exposure to air pollutants. Second, the 5-year pollutant exposure period was fixed for all participants (2008–2012) regardless of when an individual's

**Table 7. Association of carotid wall volume (mm$^3$) with PM$_{2.5}$ within co-pollutant exposure tertiles.**

| | Overall N = 6223 | | Low O$_3$ 14.2–22.9 ppb N = 1941 | | Mid O$_3$ 23.0–26.3 ppb N = 2296 | | High O$_3$ 26.3–39.1 ppb N = 1986 | |
|---|---|---|---|---|---|---|---|---|
| | Effect (95% CI) | p- | Effect (95% CI) | p- | Effect (95% CI) | p- | Effect (95% CI) | p- |
| Model 1 | -10.1 (-24.7,4.49) | 0.1747 | 5.51 (-14.6,25.63) | 0.5911 | -3.21 (-38.3,31.92) | 0.8580 | -20.5 (-40.9,-0.16) | 0.0482 |
| Model 2 | -14.9 (-28.4,-1.42) | 0.0303 | 5.89 (-19.8,31.54) | 0.6527 | -12.5 (-45.1,20.16) | 0.4533 | -21.7 (-40.5,-2.89) | 0.0238 |
| Model 3 | -15.0 (-28.5,-1.55) | 0.0288 | 6.90 (-18.8,32.64) | 0.5990 | -13.8 (-46.3,18.64) | 0.4041 | -21.9 (-40.8,-2.98) | 0.0233 |
| Model 4 | -8.40 (-23.3,6.48) | 0.2685 | 9.42 (-16.8,35.64) | 0.4809 | -15.2 (-48.8,18.41) | 0.3756 | -19.1 (-41.7,3.49) | 0.0975 |
| | | | Low NO$_2$ (0.88–8.10 ppb) N = 1539 | | Mid NO$_2$ (8.12–15.5 ppb) N = 2202 | | High NO$_2$ (15.5–38.7 ppb) N = 2482 | |
| | | | Effect (95% CI) | p- | Effect (95% CI) | p- | Effect (95% CI) | p- |
| Model 1 | -10.1 (-24.7,4.49) | 0.1747 | 24.21 (-10.3,58.67) | 0.1684 | 1.51 (-20.3,23.36) | 0.8922 | 31.73 (-4.58,68.04) | 0.0867 |
| Model 2 | -14.9 (-28.4,-1.42) | 0.0303 | 13.62 (-18.1,45.31) | 0.3995 | 0.59 (-20.2,21.34) | 0.9553 | 32.17 (-2.85,67.20) | 0.0718 |
| Model 3 | -15.0 (-28.5,-1.55) | 0.0288 | 12.68 (-19.2,44.60) | 0.4360 | 0.18 (-20.6,20.96) | 0.9862 | 33.37 (-1.54,68.27) | 0.0610 |
| Model 4 | -8.40 (-23.3,6.48) | 0.2685 | 20.92 (-13.6,55.40) | 0.2342 | -6.50 (-28.8,15.82) | 0.5678 | 35.57 (0.35,70.80) | 0.0478 |

Presented data are adjusted CWV estimates (95% CI) per 5 µg/m$^3$ increase for PM$_{2.5}$ from a linear mixed model with centre modelled as a random intercept. Model 1, unadjusted. Model 2, adjusted for age, (sex), and ethnicity. Model 3 further adjusted for the INTERHEART risk score, education and workplace location. Primary model 4 further adjusted for walkability categories, material factor score and social factor score.

**Table 8. Association of carotid wall volume (mm$^3$) with NO$_2$ within co-pollutant exposure tertiles.**

| | Overall N = 6223 | | Low O$_3$ 14.2–22.9 ppb N = 1941 | | Mid O$_3$ 23.0–26.3 ppb N = 2296 | | High O$_3$ 26.3–39.1 ppb N = 1986 | |
|---|---|---|---|---|---|---|---|---|
| | Effect (95% CI) | p- | Effect (95% CI) | p- | Effect (95% CI) | p- | Effect (95% CI) | p- |
| Model 1 | -10.1 (-14.3,-5.95) | <.0001 | 0.44 (-5.33,6.21) | 0.8821 | -4.72 (-13.4,3.95) | 0.2857 | -27.5 (-35.1,-20.0) | <.0001 |
| Model 2 | -11.2 (-15.0,-7.32) | <.0001 | 6.95 (-1.29,15.18) | 0.0981 | -6.57 (-14.5,1.38) | 0.1054 | -27.7 (-34.6,-20.8) | <.0001 |
| Model 3 | -11.0 (-14.9,-7.10) | <.0001 | 7.06 (-1.19,15.31) | 0.0935 | -5.80 (-13.7,2.13) | 0.1514 | -28.0 (-34.9,-21.0) | <.0001 |
| Model 4 | -11.8 (-16.2,-7.31) | <.0001 | 7.65 (-1.49,16.78) | 0.1008 | -7.79 (-16.2,0.65) | 0.0706 | -31.8 (-39.6,-23.9) | <.0001 |

| | Low PM$_{2.5}$ (1.78–5.64 µg/m$^3$) N = 1903 | | Mid PM$_{2.5}$ 5.65–8.18 µg/m$^3$ N = 1898 | | High PM$_{2.5}$ 8.18–11.2 µg/m$^3$ N = 2422 | |
|---|---|---|---|---|---|---|
| | Effect (95% CI) | p- | Effect (95% CI) | p- | Effect (95% CI) | p- |
| Model 1 | -10.1 (-14.3,-5.95) | <.0001 | -5.36 (-14.4,3.68) | 0.2450 | -13.5 (-23.3,-3.61) | 0.0074 | -14.8 (-22.3,-7.29) | 0.0001 |
| Model 2 | -11.2 (-15.0,-7.32) | <.0001 | -7.41 (-15.8,0.96) | 0.0828 | -11.1 (-20.7,-1.61) | 0.0219 | -15.4 (-22.3,-8.59) | <.0001 |
| Model 3 | -11.0 (-14.9,-7.10) | <.0001 | -7.68 (-16.1,0.77) | 0.0749 | -11.1 (-20.6,-1.46) | 0.0239 | -14.7 (-21.6,-7.73) | <.0001 |
| Model 4 | -11.8 (-16.2,-7.31) | <.0001 | -8.00 (-18.0,2.01) | 0.1173 | -11.8 (-22.2,-1.26) | 0.0281 | -15.3 (-22.5,-8.18) | <.0001 |

Presented data are adjusted CWV estimates (95% CI) per 5 ppb increase for NO$_2$ from a linear mixed model with centre modelled as a random intercept. Model 1, unadjusted. Model 2, adjusted for age, (sex), and ethnicity. Model 3 further adjusted for the INTERHEART risk score, education and workplace location. Primary model 4 further adjusted for walkability categories, material factor score and social factor score.

enrolment occurred in the 2014–2018 window. Thus, the exposure window was not consistently 5-years prior to enrolment for all study participants (i.e., depending on the date of participant MRI scan, the 5-year window may lag behind the MRI by ~2 years if it was done in 2014, but by up to ~6 years if it was done in 2018), which further increases risk for exposure misclassification. However, studies have demonstrated temporal stability in the spatial patterns of air pollutants over 10 years, thus temporal variability in the exposure window relative to the enrollment date is not a significant source of uncertainty [36]. Because the causally relevant window for air pollution exposures remains unknown [37], future investigations are needed to examine varying exposure time-windows and lag periods. Moreover, the spatial resolution of O$_3$ is 10 km, which may not be fine enough to capture exposure variability, may have led to a spurious association between O$_3$ and CWV, or may have been confounded by suburban living. Third, because CAHHM is a prospective pan-Canadian cohort of cohorts across five provinces and participants were selected from existing cohorts, the sample is not a random sample of the Canadian population distribution, thereby limiting the generalizability of these findings. When compared to a cohort of adults who responded to the 2015 Canadian Community Health Survey, CAHHM participants were older, of higher socioeconomic status, but had a similar mean INTERHEART risk score [38]. This does not affect the exposure-to-outcome reliability of our results within CAHHM, but generalizability to younger populations and Canadians living outside major Canadian cities should be done with caution. Fourth, because of the small number of events in our cohort (n = 156 events (2.35%)), we were not powered to look at intraplaque hemorrhage. Lastly, as with any observational study, the risk of residual confounding (from factors such as diet, lifestyle factors, or pre-existing health conditions) cannot be excluded.

## Conclusion

In healthy adults living in clean or only mildly polluted environments, we found no consistent association between air pollution and atherosclerosis. Exposure to $NO_2$ was negatively associated, and $O_3$ was positively associated with CWV, a sensitive measure of subclinical atherosclerosis, while $PM_{2.5}$ was not associated with CWV. These inconsistent results raise questions as to whether previous reports linking low-level exposure to air pollution and CVD morbidity may have suffered from uncontrolled confounding. The role of $NO_2$ in atherosclerosis is complex and requires further investigation, as do combinations of exposure to air pollutants.

## Supporting information

**S1 Fig. Flow chart for Air pollution and MRI markers in CAHHM.**
(PDF)

**S2 Fig. Scatterplot of air pollutants by carotid wall volume measurements with regression lines stratified by sex.**
(PDF)

**S1 Table. Anthropometric characteristics of the study population by sex.**
(PDF)

**S2 Table. Demographics & lifestyle characteristics of the study population by sex.**
(PDF)

**S3 Table. Environmental characteristics of the study population by sex.**
(PDF)

## Acknowledgments

**Steering Committee of Canadian Alliance of Healthy Hearts and Minds:** S.S. Anand (Chair)*, M.G. Friedrich (Co-Chair), Douglas S. Lee (Co-Chair), P Awadalla (Ontario Health Study), T. Dummer (BC Generations Project), J. Vena (Alberta's Tomorrow Project), P. Broët (CARTaGENE), J. Hicks (Atlantic PATH), J-C. Tardif (MHI Biobank), K. Teo, S. Yusuf (PURE-Central), B-M. Knoppers (Ethics, Legal and Social Issues). **Project Office Staff at Population Health Research Institute (PHRI):** D. Desai, S. Zafar. **Statistics/Biometrics Programmers Team at PHRI:** K. Schulze, S. Bangdiwala. **Cohort Operations Research Personnel:** K McDonald (Ontario Health Study), N. Noisel (CARTaGENE), J. Chu (BC Generations Project), J. Hicks (Atlantic PATH), H. Whelan (Alberta's Tomorrow Project), S. Rangarajan (PURE), D. Busseuil (MHI Biobank). **Site Investigators and Staff:** J. Leipsic, S. Lear, V. de Jong; M. Noseworthy, K. Teo, E. Ramezani, N. Konyer; P. Poirier, A-S. Bourlaud, E Larose, K. Bibeau; J. Leipsic, S. Lear, V. de Jong; E. Smith, R. Frayne, A. Charlton, R Sekhon; A. Moody, V. Thayalasuthan; A.Kripalani, G Leung; M. Noseworthy, S. Anand, R. de Souza, N. Konyer, S. Zafar; G. Paraga,L. Reid; A.J. Dick, F. Ahmad; D. Kelton, H. Shah; F. Marcotte, H. Poiffaut; M.G. Friedrich, J. Lebel; E. Larose, K. Bibeau; R. Miller, L. Parker, D. Thompson, J. Hicks; J-C. Tardif, H.Poiffaut; J. Tu, K. Chan, A. Moody, V. Thayalasuthan. **MRI Working Group and Core Lab Investigators/Staff:** Chair: M.G. Friedrich; Brain Core Lab: E.E. Smith; Carotid Core Lab: A. Moody, V. Thayalasuthan; Abdomen: E. Larose, K. Bibeau, Cardiac: F. Marcotte, F. Henriques. **Contextual Working Group:** R. de Souza, S. Anand, G. Booth, J. Brook, D. Corsi, L. Gauvin, S. Lear, F. Razak, S.V. Subramanian, J. Tu. **CAHHM Founding Advisory Group:** Jean Rouleau, Pierre Boyle, Caroline Wong, Eldon Smith.

The Canadian Alliance of Healthy Hearts and Minds (CAHHM) investigators include the following individuals: Sonia S. Anand, MD, PhD, Department of Medicine, Department of Health Research Methods, Evidence, and Impact, McMaster University, Population Health Research Institute, Hamilton Health Sciences, Hamilton, Canada; Philip Awadalla, PhD, Department of Molecular Genetics, Ontario Institute for Cancer Research, University of Toronto, Toronto, Canada; Sandra E. Black, MA, MD, OC, Department of Medicine (Neurology), Sunnybrook Health Sciences Centre, University of Toronto, Toronto, Canada; Broët Philippe, MD, PhD, Department of Preventive and Social Medicine, École de santé publique, Université de Montréal, and Research Centre, CHU Sainte Justine, Montréal, Canada; Alexander Dick, MD, Department of Medicine, University of Ottawa Heart Institute, Ottawa, Canada; Trevor Dummer, PhD, MSc, Department of Epidemiology, Biostatistics, and Public HealthPractice, School of Population and Public Health, University of British Columbia, and BC Cancer Agency, Vancouver, Canada; Matthias G. Friedrich, MD, Department of Cardiology, McGill University, Montréal, Canada; Jason Hicks, MSc, Atlantic Partnership for Tomorrow's Health, Dalhousie University, Halifax, Canada; David Kelton, MD, RPVI, Department of Medicine, William Osler Health System; Anish Kirpalani, MD, MASc, Department of Medical Imaging, St. Michael's Hospital, University of Toronto, Toronto, Canada; Maria Bartha Knoppers, PhD, Centre of Genomics and Policy, McGill University, Montréal, Canada; Scott A. Lear, PhD, Department of Pathology, Simon Fraser University, Burnaby, Canada; Eric Larose, DVM, MD, Department of Medicine, University of Laval, Quebec City, Canada; Russell J. de Souza, RD, ScD, Department of Health Research Methods, Evidence, and Impact, McMaster University, Hamilton, Canada; Douglas S. Lee, MD, PhD, ICES Central, Cardiovascular Research Program, Institute for Clinical Evaluative Sciences, Peter Munk Cardiac Centre University Health Network, Department of Medicine, University of Toronto, Toronto, Canada; Jonathan Leipsic, MD, Department of Radiology, University of British Columbia, Vancouver, Canada; Francois Marcotte, MD, Department of Cardiology, Montreal Heart Institute, University of Montreal, Montréal, Canada; Alan R. Moody, MBBS, Department of Medical Imaging, University of Toronto, Sunnybrook Health Sciences Centre, Toronto, Canada; Michael D. Noseworthy, PhD, PEng, Department of Electrical and Computer Engineering, McMaster University, St. Joseph's Health Care, Hamilton, Canada; Grace Parraga, PhD, Department of Medical Biophysics, Robarts Research Institute, Western University, London, Canada; Louise Parker, PhD, Atlantic Partnership for Tomorrow's Health, Dalhousie University, Halifax, Canada; Paul Poirier, MD, PhD, Institut de cardiologie et de pneumologie de Quebec, Université of Laval, Quebec City, Canada; Eric E. Smith, MD, MPH, Hotchkiss Brain Institute, Department of Clinical Neurosciences, University of Calgary, Calgary, Canada; Jean-Claude Tardif, MD, Department of Cardiology, Montreal Heart Institute, University of Montreal, Montréal, Canada; Koon K. Teo, MBBCh, PhD, Department of Medicine, Department of Health Research Methods, Evidence, and Impact, McMaster University, Population Health Research Institute, Hamilton Health Sciences, Hamilton, Canada; Jack V. Tu, MD, PhD, MSc, Department of Medicine, University of Toronto, Institute for Clinical Evaluative Sciences, Sunnybrook Schulich Heart Centre, Toronto, Canada (deceased); Jennifer Vena, PhD, Cancer Research and Analytics, Cancer Control Alberta, Alberta Health Services, Edmonton, Canada.

## Author Contributions

**Conceptualization:** Jeffrey R. Brook, Russell J. de Souza.

**Data curation:** Karleen M. Schulze.

**Formal analysis:** Sandi M. Azab, Karleen M. Schulze, Jeffrey R. Brook, Russell J. de Souza.

**Investigation:** Michael Brauer, Eric E. Smith, Matthias G. Friedrich, Douglas Lee, Trevor J. B. Dummer, Paul Poirier, Jean-Claude Tardif, Koon K. Teo, Scott Lear, Salim Yusuf, Sonia S. Anand.

**Methodology:** Dany Doiron, Karleen M. Schulze, Jeffrey R. Brook, Alan R. Moody, Matthias G. Friedrich, Shrikant I. Bangdiwala, Sonia S. Anand, Russell J. de Souza.

**Project administration:** Dipika Desai.

**Supervision:** Jeffrey R. Brook, Sonia S. Anand, Russell J. de Souza.

**Visualization:** Sandi M. Azab.

**Writing – original draft:** Sandi M. Azab, Sonia S. Anand, Russell J. de Souza.

**Writing – review & editing:** Sandi M. Azab, Dany Doiron, Karleen M. Schulze, Jeffrey R. Brook, Michael Brauer, Eric E. Smith, Alan R. Moody, Dipika Desai, Matthias G. Friedrich, Shrikant I. Bangdiwala, Dena Zeraatkar, Douglas Lee, Trevor J. B. Dummer, Paul Poirier, Jean-Claude Tardif, Koon K. Teo, Scott Lear, Salim Yusuf, Sonia S. Anand, Russell J. de Souza.

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
