## [Decision Letter · Decision Letter 0]

22 Aug 2023

PONE-D-23-23654Ambient Air Pollution and Subclinical Carotid Atherosclerosis Measured by Magnetic Resonance Imaging: A Prospective Cohort Study

PLOS ONE

Dear Dr. de Souza,

Thank you for submitting your manuscript to PLOS ONE. After careful consideration, we feel that it has merit but does not fully meet PLOS ONE’s publication criteria as it currently stands. Therefore, we invite you to submit a revised version of the manuscript that addresses the points raised during the review process.

Dear Authors,

I have reviewed your manuscript detailing the association between air pollution and subclinical atherosclerosis in a Canadian cohort. Below, I outline both the strengths of your work and areas that require attention and revision.

Strengths:

Subject Matter: The topic of your study is highly relevant and timely, addressing an important public health concern.

Methodology: The use of MRI-characterized CWV to assess atherosclerosis is innovative and provides a more precise measurement.

Please address reviewer comments made by the two reviewers.

Furthermore, Areas for Revision:

Exposure Misclassification: Please provide a more detailed explanation of how individual-level exposures were estimated based on residence address. Consideration of participants' time away from residence and residential history could enhance the accuracy of the exposure assessment.

Inconsistent Exposure Window: The fixed 5-year pollutant exposure period for all participants may introduce bias. Please discuss this limitation and consider conducting sensitivity analyses to assess its impact.

Unexpected Findings: The negative association of NO2 with CWV and the lack of association with PM2.5 were unexpected. A more comprehensive discussion of these findings and potential underlying factors is needed.

Resolution of O3 Data: The spatial resolution of O3 data (10 km) may not be fine enough to capture local variations. Please discuss this limitation and its potential impact on the findings related to O3.

Lack of Consideration for Intraplaque Hemorrhage: Please discuss the limitation of not being able to look at intraplaque hemorrhage and its potential significance in understanding the relationship between air pollution and atherosclerosis.

Additional Limitations: Consider addressing other potential limitations such as selection bias, lack of longitudinal data, potential confounding variables, measurement error in pollution data, limited geographic scope, and potential interaction effects.

Your study has the potential to make a significant contribution to the field. However, the above concerns need to be addressed to enhance the robustness and credibility of the findings. I recommend a minor revision that includes a more detailed analysis of the exposure assessment, consideration of additional confounding variables, and a more comprehensive discussion of the unexpected findings and other limitations.

I look forward to seeing the revised manuscript.

We look forward to receiving your revised manuscript.

Kind regards,

Sabeena Jalal, MBBS, MSc, MSc, SM

Academic Editor

PLOS ONE

Journal Requirements:

"I have read the journal's policy and the authors of this manuscript have the following competing interests:

RJ de Souza has served as an external resource person to the World Health Organization’s Nutrition Guidelines Advisory Group on trans fats, saturated fats, and polyunsaturated fats.  The WHO paid for his travel and accommodation to attend meetings from 2012-2017 to present and discuss this work.  He has presented updates of this work to the WHO in 2022. He has also done contract research for the Canadian Institutes of Health Research’s Institute of Nutrition, Metabolism, and Diabetes, Health Canada, and the World Health Organization for which he received remuneration.  He has received speaker’s fees from the University of Toronto, and McMaster Children’s Hospital. He has held grants from the Canadian Institutes of Health Research, Canadian Foundation for Dietetic Research, Population Health Research Institute, and Hamilton Health Sciences Corporation as a principal investigator, and is a co-investigator on several funded team grants from the Canadian Institutes of Health Research. He has served as an independent director of the Helderleigh Foundation (Canada). He serves as a member of the Nutrition Science Advisory Committee to Health Canada (Government of Canada), and a co-opted member of the Scientific Advisory Committee on Nutrition (SACN) Subgroup on the Framework for the Evaluation of Evidence (Public Health England). Dr Anand reported receiving grants from Canadian Partnership Against Cancer, Heart and Stroke Foundation of Canada, and Canadian Institutes of Health Research, and a Canadian Institutes of Health Research Foundation grant during the conduct of the study and serving as the Tier 1 Canada Research Chair Ethnicity and Cardiovascular Disease and as the Michael G Degroote Heart and Stroke Foundation Chair in Population Health Research, and receiving grants from Heart and Stroke Foundation of Canada and Canadian Institutes of Health Research, and receiving personal fees from Bayer outside the submitted work. Dr Friedrich reported receiving personal fees from Circle CVI Inc for serving as a board member and adviser and being a shareholder outside the submitted work. Dr Dummer reported receiving grants from Canadian Partnership Against Cancer during the conduct of the study. Dr Lear reported receiving grants from the Canadian Institutes of Health Research and grants from Michael Smith Foundation for Health Research during the conduct of the study and personal fees from Curatio Inc outside the submitted work. Dr Tardif reported receiving grants from Amarin, Ceapro, Esperion, Ionis, Novartis, Pfizer, RegenXBio, Sanofi, AstraZeneca, and DalCor Pharmaceuticals, receiving personal fees from AstraZeneca, HLS Pharmaceuticals, Pendopharm, and DalCor Pharmaceuticals, and having a minor equity interest in DalCor Pharmaceuticals Minor outside the submitted work. In addition, Dr Tardif had a patent for Pharmacogenomics-Guided CETP Inhibition issued by DalCor Pharmaceuticals, a patent for Use of Colchicine After Myocardial Infarction pending, and a patent for Genetic Determinants of Response to Colchicine pending. No other disclosures were reported. Dr Brauer served on the WHO Guideline Development Group (no remuneration was provided but travel costs to meetings were covered). "

Additional Editor Comments:

Dear Authors,

I have reviewed your manuscript detailing the association between air pollution and subclinical atherosclerosis in a Canadian cohort. Below, I outline both the strengths of your work and areas that require attention and revision.

Strengths:

Subject Matter: The topic of your study is highly relevant and timely, addressing an important public health concern.

Methodology: The use of MRI-characterized CWV to assess atherosclerosis is innovative and provides a more precise measurement.

Areas for Revision:

Exposure Misclassification: Please provide a more detailed explanation of how individual-level exposures were estimated based on residence address. Consideration of participants' time away from residence and residential history could enhance the accuracy of the exposure assessment.

Inconsistent Exposure Window: The fixed 5-year pollutant exposure period for all participants may introduce bias. Please discuss this limitation and consider conducting sensitivity analyses to assess its impact.

Unexpected Findings: The negative association of NO2 with CWV and the lack of association with PM2.5 were unexpected. A more comprehensive discussion of these findings and potential underlying factors is needed.

Resolution of O3 Data: The spatial resolution of O3 data (10 km) may not be fine enough to capture local variations. Please discuss this limitation and its potential impact on the findings related to O3.

Lack of Consideration for Intraplaque Hemorrhage: Please discuss the limitation of not being able to look at intraplaque hemorrhage and its potential significance in understanding the relationship between air pollution and atherosclerosis.

Additional Limitations: Consider addressing other potential limitations such as selection bias, lack of longitudinal data, potential confounding variables, measurement error in pollution data, limited geographic scope, and potential interaction effects.

Conclusion:

Your study has the potential to make a significant contribution to the field. However, the above concerns need to be addressed to enhance the robustness and credibility of the findings. I recommend a minor revision that includes a more detailed analysis of the exposure assessment, consideration of additional confounding variables, and a more comprehensive discussion of the unexpected findings and other limitations.

I look forward to seeing the revised manuscript.

Thank you.

Reviewers' comments:

Reviewer's Responses to Questions

**Comments to the Author**

1. Is the manuscript technically sound, and do the data support the conclusions?

Reviewer #1: Partly

Reviewer #2: Partly

2. Has the statistical analysis been performed appropriately and rigorously? 

Reviewer #1: Yes

Reviewer #2: Yes

3. Have the authors made all data underlying the findings in their manuscript fully available?

Reviewer #1: Yes

Reviewer #2: Yes

4. Is the manuscript presented in an intelligible fashion and written in standard English?

Reviewer #1: Yes

Reviewer #2: Yes

5. Review Comments to the Author

Reviewer #1: Hello,

The study presents valuable insights into the relationship between air pollution and subclinical atherosclerosis, utilizing a large sample size and advanced MRI techniques. The geographical diversity and comprehensive analysis add to the study's strengths.

However, there are areas that require clarification and improvement:

Exposure Assessment: The authors should provide a more detailed explanation of the exposure misclassification risk and how the fixed 5-year pollutant exposure period might have affected the results. Clarifying these aspects would enhance the paper's transparency and allow readers to better assess the findings' validity.

Unexpected Findings: The unexpected negative association with NO2 and the lack of association with PM2.5 should be discussed more thoroughly. The authors should explore potential reasons for these findings and compare them more extensively with existing literature.

Resolution of O3 Data: The authors should discuss the limitations of the O3 spatial resolution and how it might have influenced the results.

Additional Analyses (Optional): If possible, further analyses could be conducted to address some of the limitations, such as considering participants' time away from residence or exploring the relationship with intraplaque hemorrhage.

The paper's contributions are significant, and the limitations do not undermine the overall value of the research. However, addressing these minor revisions would enhance the paper's clarity, coherence, and impact. Therefore, acceptance with minor revisions seems the most appropriate recommendation.

Thank you.

Reviewer #2: Mention the limitations

Selection Bias: If the study's participants were not randomly selected or if there was a lack of diversity in the sample (e.g., age, gender, geographic location), this could introduce selection bias, limiting the generalizability of the findings.

Lack of Longitudinal Data: If the study was cross-sectional in nature, the lack of longitudinal data might hinder the ability to establish causal relationships between air pollution and atherosclerosis.

Potential Confounding Variables: If not all relevant confounding variables were controlled for, such as diet, lifestyle factors, or pre-existing health conditions, this could affect the validity of the associations found.

Measurement Error in Pollution Data: If there were inaccuracies in the measurement of pollution levels, such as reliance on satellite data or modeling without sufficient ground-truthing, this could lead to misestimation of exposure levels.

Lack of Sensitivity Analysis: If sensitivity analyses were not conducted to assess the robustness of the findings to different modeling assumptions or potential outliers, this could raise questions about the stability of the results.

Concerns Regarding Exposure Assessment: The methodology used to gauge individual-level exposures, relying solely on residence addresses, might lead to inaccuracies in exposure classification. The failure to account for variations in participants' locations and their residential histories could compromise the integrity of the exposure evaluation.

Inconsistency in Exposure Time Frame: The application of a uniform 5-year window for pollutant exposure across all participants, irrespective of their specific enrollment dates, could introduce an element of bias, potentially skewing the assessment of exposure.

Unanticipated Results: The findings that NO2 was inversely associated with CWV and that there was no discernible connection with PM2.5 were surprising and at odds with certain existing studies. These outcomes prompt questions about possible unaccounted confounding variables or other hidden factors that might have shaped the results.

Limitations in O3 Data Resolution: The 10 km spatial resolution used for O3 data might be insufficient to detect localized fluctuations, which could have an impact on the precision of the conclusions drawn regarding O3.

Omission of Intraplaque Hemorrhage Analysis: The study's inability to examine intraplaque hemorrhage, owing to its limited scope, might overlook a crucial aspect in unraveling the complex relationship between air pollution and atherosclerosis. This limitation could be significant in the overall interpretation of the findings.

6. PLOS authors have the option to publish the peer review history of their article (what does this mean?). If published, this will include your full peer review and any attached files.

Reviewer #1: No

Reviewer #2: No

---

## [Author Response · Author response to Decision Letter 0]

17 Oct 2023

3 October, 2023

Sabeena Jalal, MBBS, MSc, MSc, SM

Academic Editor

PLOS ONE

Response to initial response: [PONE-D-23-23654] ‘Ambient Air Pollution and Subclinical Carotid Atherosclerosis Measured by Magnetic Resonance Imaging: A Prospective Cohort Study’

Dear Dr. Jalal,

We are pleased to resubmit our manuscript to PLOS ONE following the 2nd review. We have edited the manuscript and provide a point-by-point response to your comments. We feel based on your comments, and those from reviewers, that the manuscript quality has been strengthened and we look forward to your second review and response. 

Please note that the editorial and reviewers’ comments are in blue italics, while our responses follow in normal font. We have tracked all changes made to the revised manuscript based on the reviewers’ comments in one copy of the manuscript.

Sincerely,

Russell de Souza, Sc.D., R.D.

Associate Professor, Population Genomics Program

Department of Health Research Methods, Evidence, and Impact

1200 Main St. West 

MDCL, Room 3210

Hamilton, Ontario, Canada 

L8N 3Z5 

Phone (905) 525-9140 x. 22109

desouzrj@mcmaster.ca

Editor:

Dear Authors,

I have reviewed your manuscript detailing the association between air pollution and subclinical atherosclerosis in a Canadian cohort. Below, I outline both the strengths of your work and areas that require attention and revision.

Strengths:

Subject Matter: The topic of your study is highly relevant and timely, addressing an important public health concern.

Methodology: The use of MRI-characterized CWV to assess atherosclerosis is innovative and provides a more precise measurement.

Please address reviewer comments made by the two reviewers.

Furthermore, Areas for Revision:

Exposure Misclassification: Please provide a more detailed explanation of how individual-level exposures were estimated based on residence address. Consideration of participants' time away from residence and residential history could enhance the accuracy of the exposure assessment.

Inconsistent Exposure Window: The fixed 5-year pollutant exposure period for all participants may introduce bias. Please discuss this limitation and consider conducting sensitivity analyses to assess its impact.

Unexpected Findings: The negative association of NO2 with CWV and the lack of association with PM2.5 were unexpected. A more comprehensive discussion of these findings and potential underlying factors is needed.

Resolution of O3 Data: The spatial resolution of O3 data (10 km) may not be fine enough to capture local variations. Please discuss this limitation and its potential impact on the findings related to O3.

Lack of Consideration for Intraplaque Hemorrhage: Please discuss the limitation of not being able to look at intraplaque hemorrhage and its potential significance in understanding the relationship between air pollution and atherosclerosis.

Additional Limitations: Consider addressing other potential limitations such as selection bias, lack of longitudinal data, potential confounding variables, measurement error in pollution data, limited geographic scope, and potential interaction effects.

Your study has the potential to make a significant contribution to the field. However, the above concerns need to be addressed to enhance the robustness and credibility of the findings. I recommend a minor revision that includes a more detailed analysis of the exposure assessment, consideration of additional confounding variables, and a more comprehensive discussion of the unexpected findings and other limitations.

We thank the Editor for recognizing the value of our work in cardiovascular disease and for sharing the reviewers’ viewpoints on our manuscript. We are confident that the raised points are adequately addressed in the revisions we have made as well as our detailed point-by-point responses below.

Reviewer #1: 

Hello,

The study presents valuable insights into the relationship between air pollution and subclinical atherosclerosis, utilizing a large sample size and advanced MRI techniques. The geographical diversity and comprehensive analysis add to the study's strengths. However, there are areas that require clarification and improvement:

We thank Reviewer 1 for taking the time to evaluate our manuscript, for the favourable response, and for the constructive feedback.

1. Exposure Assessment: The authors should provide a more detailed explanation of the exposure misclassification risk and how the fixed 5-year pollutant exposure period might have affected the results. Clarifying these aspects would enhance the paper's transparency and allow readers to better assess the findings' validity.

We thank Reviewer 1 for pointing out the need for further clarity, a point also raised by Reviewer 2, that we shall address collectively here. We have now further elaborated on this limitation (Discussion, page 19) adding that “the exposure window was not consistently 5-years prior to enrolment for all study participants (i.e., depending on the date of participant MRI scan, the 5 year window may lag behind the MRI by ~2 years if it was done in 2014, but by up to ~6 years if it was done in 2018), which further increases risk for exposure misclassification” and that “future investigations are needed to examine varying exposure time-windows and lag periods.”

2. Unexpected Findings: The unexpected negative association with NO2 and the lack of association with PM2.5 should be discussed more thoroughly. The authors should explore potential reasons for these findings and compare them more extensively with existing literature.

We agree with both Reviewer 1 and 2 that the negative association was surprising and unexpected. It is quite plausible that air pollution would not adversely affect all biological systems. Other MRI-measured outcome, including ectopic adipose tissue, silent vascular brain injury and brain volume, carotid atherosclerosis, left ventricular volumes, function and mass and silent myocardial infarction (MI) may not be associated with air pollution. We are currently examining some of these additional outcomes in work in progress. We have further discussed this point: “Collectively within the existing body of literature, it is plausible that NO2 is probably not involved in early carotid thickening but perhaps in more advanced morbid stages.” 

We have also compared our NO2 results more extensively with existing literature (Discussion, page 18): “MESA-Air did not find a relationship between NO2 or other pollutant exposures and IMT change, instead exposure was positively associated with coronary artery calcification progression .2 In four European cohort, ESCAPE findings were inconsistent for an association between NO2 and cIMT, in fact, all four cohorts and their meta-analytical estimate, showed an inverse association, similar to our observation in CAHHM.” Please find below ESCAPE results summarized in Figure 1 of the paper.

Figure 1. Forest plot of the percent difference in CIMT (geometric mean with 95% CIs) for model M3 for (A) ESCAPE air pollutants per standard contrast of exposure as indicated in the figure. Source: Environmental Health Perspectives • volume 123 | number 6 | June 2015

3. Resolution of O3 Data: The authors should discuss the limitations of the O3 spatial resolution and how it might have influenced the results.

We agree with Reviewer 1 that the resolution of O3 data is a limitation of our study. We have stated this limitation under methods and twice in the discussion. However, the same O3 data has been used in the MAPLE study (the major, recent air pollution and mortality work in Canada), where O3 was seen to be impacting mortality. Nevertheless, we have now added this limitation in the abstract and again in the last paragraph of the discussion (page 20), acknowledging the potential for a spurious association due to the difference in spatial resolution of this marker. More-resolved exposure estimates of O3 are very much needed for future work to better understand this, but this is complex because of the strong inverse correlation between O3 and NO2 as you go to finer and finer resolutions.

4. Additional Analyses (Optional): If possible, further analyses could be conducted to address some of the limitations, such as considering participants' time away from residence or exploring the relationship with intraplaque hemorrhage.

We thank Reviewer 1 for this suggestion. We would like to point out that we have adjusted in our models for residing at home/working in the lived-in community versus working outside the lived-in community. We have now added to our sensitivity analyses, the stratified analysis based on this variable as a proxy for time away from residential address. These sensitivity analyses showed results consistent with our main analysis for all three pollutants for those working in the community (supplemental Table S5). As for intraplaque hemorrhage, we acknowledge that it is an important clinical/atherosclerosis outcome to investigate, however, in this cohort, prevalence of IPH (n=156 IPH events (2.35%)) is too low to be able to estimate this association reliably.

5. The paper's contributions are significant, and the limitations do not undermine the overall value of the research. However, addressing these minor revisions would enhance the paper's clarity, coherence, and impact. Therefore, acceptance with minor revisions seems the most appropriate recommendation. Thank you.

We would like to thank Reviewer 1 for recognising the value of our work within the field. We feel based on your comments that the manuscript quality has been strengthened and we look forward to your second review and response.

Reviewer #2: 

Mention the limitations. 

1. Selection Bias: If the study's participants were not randomly selected or if there was a lack of diversity in the sample (e.g., age, gender, geographic location), this could introduce selection bias, limiting the generalizability of the findings.

We thank Reviewer 2 for this comment. We have now added this limitation to the final paragraph of the discussion: “Third, because CAHHM is a prospective pan-Canadian cohort of cohorts across five provinces and participants were selected from existing cohorts, the sample is not a random sample of the Canadian population distribution, thereby limiting the generalizability of these findings. When compared to a cohort of adults who responded to the 2015 Canadian Community Health Survey, CAHHM participants were older, of higher socioeconomic status, but had a similar mean INTERHEART risk score.38 This does not affect the exposure-to-outcome reliability of our results within CAHHM, but generalizability to younger populations and Canadians living outside major Canadian cities should be done with caution.”

2. Lack of Longitudinal Data: If the study was cross-sectional in nature, the lack of longitudinal data might hinder the ability to establish causal relationships between air pollution and atherosclerosis.

We have elaborated on this limitation based on the Bradford Hill criteria for causality. “Exposure to air pollution values represented a time frame prior to knowledge of the outcome for each participant, i.e., the air pollution data collected for the 5-year period prior to the MRI. We are therefore comfortable describing this as a prospective association, despite the lack of longitudinal follow-up.”

3. Potential Confounding Variables: If not all relevant confounding variables were controlled for, such as diet, lifestyle factors, or pre-existing health conditions, this could affect the validity of the associations found.

We thank Reviewer 2 for this point of consideration. We argue that a major strength of our study is the detailed and well-characterized phenotyping of our cohort. We have considered a parsimonious yet comprehensive set of confounders based on subject matter expertise and prior literature. These include the INTERHEART risk score that captures some dietary components, physical exercise, lifestyle and comorbidities summarizing individual cardiovascular risk. Nevertheless, we have now added to the limitations (Discussion, page 20) that “as with any observational study, the risk of residual confounding (from factors such as diet, lifestyle factors, or pre-existing health conditions) cannot be excluded.”

4. Measurement Error in Pollution Data: If there were inaccuracies in the measurement of pollution levels, such as reliance on satellite data or modeling without sufficient ground-truthing, this could lead to misestimation of exposure levels.

We thank Reviewer 2 for this comment, however, as we mention under (Methods, page 7), “To adjust for any residual bias in the satellite-derived PM2.5 estimates, a geographically weighted regression (GWR) incorporating ground-based observations was then applied. Good agreement was found with cross-validated surface observations across North-America (R2 = 0.70).” This geographically-weighted regression uses the surface (ground-truth) PM2.5 to correct for bias in the satellite estimates resulting in an improved R2 so that the exposures (at least at the scale of North America as a whole) are not biased and the magnitude of the gradient (from lowest to highest exposure) is representative of what ground measurements show. Inevitably, there are limited numbers of ground stations, thus, there is still room for errors but we are using the best available estimates at this time.

5. Lack of Sensitivity Analysis: If sensitivity analyses were not conducted to assess the robustness of the findings to different modeling assumptions or potential outliers, this could raise questions about the stability of the results.

We thank Reviewer 2 for this comment. Sensitivity analyses were conducted as outlined under statistical analysis in methods (results in supplemental table S4). In sensitivity analyses, models 1-5 were i. stratified by sex and ii. repeated after excluding participants based on immigration status for those who had been in Canada for less than ten years (n=5885). Moreover, as mentioned in our response to Reviewer 1, we have now added to our sensitivity analyses, iii. a stratified analysis based on workplace location (working outside the community) to address possible exposure misclassification.

6. Concerns Regarding Exposure Assessment: The methodology used to gauge individual-level exposures, relying solely on residence addresses, might lead to inaccuracies in exposure classification. The failure to account for variations in participants' locations and their residential histories could compromise the integrity of the exposure evaluation.

We have addressed this concern in our response to Reviewer 1; please see above under point #1.

7. Inconsistency in Exposure Time Frame: The application of a uniform 5-year window for pollutant exposure across all participants, irrespective of their specific enrollment dates, could introduce an element of bias, potentially skewing the assessment of exposure.

Thank you. We have addressed this limitation above under point #1 and strived to make it clearer and more transparent in the manuscript.

8. Unanticipated Results: The findings that NO2 was inversely associated with CWV and that there was no discernible connection with PM2.5 were surprising and at odds with certain existing studies. These outcomes prompt questions about possible unaccounted confounding variables or other hidden factors that might have shaped the results.

We have addressed this concern in our response to Reviewer 1; please see above under point #2.

9. Limitations in O3 Data Resolution: The 10 km spatial resolution used for O3 data might be insufficient to detect localized fluctuations, which could have an impact on the precision of the conclusions drawn regarding O3.

We have addressed this concern in our response to Reviewer 1; please see above under point #3.

10. Omission of Intraplaque Hemorrhage Analysis: The study's inability to examine intraplaque hemorrhage, owing to its limited scope, might overlook a crucial aspect in unraveling the complex relationship between air pollution and atherosclerosis. This limitation could be significant in the overall interpretation of the findings.

We have addressed this concern in our response to Reviewer 1; please see 

---

## [Decision Letter · Decision Letter 1]

8 Mar 2024

PONE-D-23-23654R1Ambient Air Pollution and Subclinical Carotid Atherosclerosis Measured by Magnetic Resonance Imaging: A Prospective Cohort StudyPLOS ONE

Dear Dr. de Souza,

Thank you for submitting your manuscript to PLOS ONE. After careful consideration, we feel that it has merit but does not fully meet PLOS ONE’s publication criteria as it currently stands. Therefore, we invite you to submit a revised version of the manuscript that addresses the points raised during the review process.

**ACADEMIC EDITOR: **

1. Although "air pollution" is cited in the study title, "air pollutants exposure" is explicitly addressed in the findings. Therefore, the study should use more precise language to avoid drawing readers' attention away from the discrepancy between the title and the findings. Suggested Title: Exposure to Air Pollutants and Subclinical Carotid Atherosclerosis Measured by Magnetic Resonance Imaging: A Prospective Cohort Study

2. Tables S4 through S8 appear to include the study's most significant findings. Given that these are the most significant findings, why are these tables listed in the supplementary materials? These should be mentioned in the main manuscript.

3. Please take into account the feedback from the reviewers and thoroughly proofread the study for errors and grammatical flaws.

4. The manuscript may be accepted after minor revisions.

We look forward to receiving your revised manuscript.

Kind regards,

Muhammad Maaz Arif, M.B.B.S, M.Phil

Academic Editor

PLOS ONE

Journal Requirements:

Additional Editor Comments (if provided):

ACADEMIC EDITOR Comments:

Following changes are the top-most priority considering the reviewers comments:

1. Although "air pollution" is cited in the study title, "air pollutants exposure" is explicitly addressed in the findings. Therefore, the study should use more precise language to avoid drawing readers' attention away from the discrepancy between the title and the findings. Suggested Title: Exposure to Air Pollutants and Subclinical Carotid Atherosclerosis Measured by Magnetic Resonance Imaging: A Prospective Cohort Study.

2. Tables S4 through S8 appear to include the study's most significant findings. Given that these are the most significant findings, why are these tables listed in the supplementary materials? These should be mentioned in the main manuscript.

3. Please take into account the feedback from the reviewers and thoroughly proofread the study for errors and grammatical flaws.

4. The manuscript may be accepted after minor revisions.

Reviewer's Comments:

The study aimed to investigate the effect of long-term exposure to air pollution on cardiovascular mortality. Therefore, the authors used data from a cohort study of 6,645 individuals recruited from previous Canadian cohort studies. The strength of this study is the well assessed outcome of carotid vessel wall volume (CWV) by MRI, the four-year average duration of air pollution exposure, and the extensive adjustment for confounders including walkability and neighborhood socioeconomic status.

Interestingly, PM2.5 and NO2 were not or negatively associated with CVW, which requires further investigation.

The reviewers' comments were understandable, and the authors responded to and addressed all comments thoroughly. Furthermore, the feedback has been extensively implemented, including additional sensitivity analyses, resulting in a substantial improvement of the manuscript, which adequately discusses the present limitations. However, there are still a few minor points that need to be addressed:

## Minor

- Abstract

o for the effect of NO2, a “minus” is missing in the 95%-CI, it must be -7.32

- Methods

o More description is needed on the exclusion criterion of "known CVD history". Is this validated or self-reported? And what diseases were included in the CVD history?

o For clarification: add a sentence describing whether residential addresses were only available on a postcode grid. If so, then PM2.5 available on a 1km*1km grid was probably averaged over the postcode area? This is not clearly described in the methods section.

o As an additional analysis, the ratio of wall volume to total vessel volume (Normalized wall index) should be used instead of using maximum wall volume [e.g. PMID: 33183741]. This would account for individual variation in overall vessel size.

- Results:

o Information in the text do not match the information in the table, e.g., in the Results - participant characteristics section it says "54.8% of participants were women", but in Table 1 it says 56%. Same section: "92% of cohort postcodes were in urban areas", but in Table 3 it is 96.2%. It seems that information in the text was mixed up with the “British Columbia” column. Authors should double-check tables and text to correct any discrepancies.

- Discussion

o Any clinically relevant cut-off values for CWV? Is the mean value high (900 mm³)? Could be more addressed in the discussion and may help to contextualize the results and health of study participants to previous studies.

o I agree with Reviewer 2 that the study was rather cross-sectional in nature than longitudinal. The authors argue “Exposure to air pollution values represented a time frame prior to knowledge of the outcome for each participant, i.e., the air pollution data collected for the 5-year period prior to the MRI. We are therefore comfortable describing this as a prospective association, despite the lack of longitudinal follow-up.” However, since no information exists on the CVW thickness before the exposure window, one cannot argue that the thickness was not present at this exposure window or even before. Therefore, I suggest revising this sentence and to name it a cross-sectional association.

- Optional: instead of A, B, C, in the plots, naming the air pollutants would improve the readability of the plots.

Reviewers' comments:

Reviewer's Responses to Questions

**Comments to the Author**

1. If the authors have adequately addressed your comments raised in a previous round of review and you feel that this manuscript is now acceptable for publication, you may indicate that here to bypass the “Comments to the Author” section, enter your conflict of interest statement in the “Confidential to Editor” section, and submit your "Accept" recommendation.

Reviewer #3: All comments have been addressed

2. Is the manuscript technically sound, and do the data support the conclusions?

Reviewer #3: Yes

3. Has the statistical analysis been performed appropriately and rigorously? 

Reviewer #3: Yes

4. Have the authors made all data underlying the findings in their manuscript fully available?

Reviewer #3: No

5. Is the manuscript presented in an intelligible fashion and written in standard English?

Reviewer #3: Yes

6. Review Comments to the Author

Reviewer #3: The study aimed to investigate the effect of long-term exposure to air pollution on cardiovascular mortality. Therefore, the authors used data from a cohort study of 6,645 individuals recruited from previous Canadian cohort studies. The strength of this study is the well assessed outcome of carotid vessel wall volume (CWV) by MRI, the four-year average duration of air pollution exposure, and the extensive adjustment for confounders including walkability and neighborhood socioeconomic status.

Interestingly, PM2.5 and NO2 were not or negatively associated with CVW, which requires further investigation.

The reviewers' comments were understandable, and the authors responded to and addressed all comments thoroughly. Furthermore, the feedback has been extensively implemented, including additional sensitivity analyses, resulting in a substantial improvement of the manuscript, which adequately discusses the present limitations. However, there are still a few minor points that need to be addressed:

## Minor

- Abstract

o for the effect of NO2, a “minus” is missing in the 95%-CI, it must be -7.32

- Methods

o More description is needed on the exclusion criterion of "known CVD history". Is this validated or self-reported? And what diseases were included in the CVD history?

o For clarification: add a sentence describing whether residential addresses were only available on a postcode grid. If so, then PM2.5 available on a 1km*1km grid was probably averaged over the postcode area? This is not clearly described in the methods section.

o As an additional analysis, the ratio of wall volume to total vessel volume (Normalized wall index) should be used instead of using maximum wall volume [e.g. PMID: 33183741]. This would account for individual variation in overall vessel size.

- Results:

o Information in the text do not match the information in the table, e.g., in the Results - participant characteristics section it says "54.8% of participants were women", but in Table 1 it says 56%. Same section: "92% of cohort postcodes were in urban areas", but in Table 3 it is 96.2%. It seems that information in the text was mixed up with the “British Columbia” column. Authors should double-check tables and text to correct any discrepancies.

- Discussion

o Any clinically relevant cut-off values for CWV? Is the mean value high (900 mm³)? Could be more addressed in the discussion and may help to contextualize the results and health of study participants to previous studies.

o I agree with Reviewer 2 that the study was rather cross-sectional in nature than longitudinal. The authors argue “Exposure to air pollution values represented a time frame prior to knowledge of the outcome for each participant, i.e., the air pollution data collected for the 5-year period prior to the MRI. We are therefore comfortable describing this as a prospective association, despite the lack of longitudinal follow-up.” However, since no information exists on the CVW thickness before the exposure window, one cannot argue that the thickness was not present at this exposure window or even before. Therefore, I suggest revising this sentence and to name it a cross-sectional association.

- Optional: instead of A, B, C, in the plots, naming the air pollutants would improve the readability of the plots.

7. PLOS authors have the option to publish the peer review history of their article (what does this mean?). If published, this will include your full peer review and any attached files.

Reviewer #3: No

---

## [Author Response · Author response to Decision Letter 1]

4 Apr 2024

ACADEMIC EDITOR Comments:

Following changes are the top-most priority considering the reviewers comments:

1. Although "air pollution" is cited in the study title, "air pollutants exposure" is explicitly addressed in the findings. Therefore, the study should use more precise language to avoid drawing readers' attention away from the discrepancy between the title and the findings. Suggested Title: Exposure to Air Pollutants and Subclinical Carotid Atherosclerosis Measured by Magnetic Resonance Imaging: A Prospective Cohort Study.

We thank the Academic Editor for this suggestion; we have changed the title accordingly.

2. Tables S4 through S8 appear to include the study's most significant findings. Given that these are the most significant findings, why are these tables listed in the supplementary materials? These should be mentioned in the main manuscript.

Table S4 is summarized in Figure 1 in the main manuscript; Table S5 is additional (secondary) sensitivity analysis while tables S6-S8 are all succinctly summarized in Figure 2. Thus, we chose to keep these tables as supplemental to avoid redundancy, while presenting the main messages of these analyses through the figures in the main manuscript. The additional detail is available for detailed review, should a reader wish, but we felt the current presentation to be easier to understand. However, if you feel very strongly, we can move these to the main text.

3. Please take into account the feedback from the reviewers and thoroughly proofread the study for errors and grammatical flaws.

We thank the Academic Editor for this comment and have now proofread and revised the article accordingly.

4. The manuscript may be accepted after minor revisions.

We thank the Academic Editor for re-reviewing our revised manuscript and for all the valuable insights to strengthen our manuscript.

Reviewer's Comments:

The study aimed to investigate the effect of long-term exposure to air pollution on cardiovascular mortality. Therefore, the authors used data from a cohort study of 6,645 individuals recruited from previous Canadian cohort studies. The strength of this study is the well assessed outcome of carotid vessel wall volume (CWV) by MRI, the four-year average duration of air pollution exposure, and the extensive adjustment for confounders including walkability and neighborhood socioeconomic status.

We thank the Reviewer for echoing the strengths and comprehensive analyses conducted within the Canadian Alliance for Healthy Hearts and Minds (CAHHM).

Interestingly, PM2.5 and NO2 were not or negatively associated with CVW, which requires further investigation.

The reviewers' comments were understandable, and the authors responded to and addressed all comments thoroughly. Furthermore, the feedback has been extensively implemented, including additional sensitivity analyses, resulting in a substantial improvement of the manuscript, which adequately discusses the present limitations. However, there are still a few minor points that need to be addressed:

We thank the Reviewer for taking the time to evaluate our revised manuscript and for the favourable response. 

## Minor

-Abstract

 for the effect of NO2, a “minus” is missing in the 95%-CI, it must be -7.32

Thank you for noting this error; this is now corrected.

- Methods

More description is needed on the exclusion criterion of "known CVD history". Is this validated or self-reported? And what diseases were included in the CVD history?

We have now added how CVD history was defined as following: (defined as a self-reported history of stroke, coronary artery disease, heart failure, or other heart disease).

o For clarification: add a sentence describing whether residential addresses were only available on a postcode grid. If so, then PM2.5 available on a 1km*1km grid was probably averaged over the postcode area? This is not clearly described in the methods section.

The following paragraph has now been added to the methods section:

“For most residential addresses, postal code areas were considerably smaller than 1x1 km so that the assigned PM2.5 concentration matches the 1x1 km grid square that the postal code is found within. Specifically, assigning PM2.5 to postal codes was performed using the single linkage approach where the PM2.5 grid square selected was the one closest to the x, y coordinate within a postal code polygon that best represents where the majority of the population lived.”

o As an additional analysis, the ratio of wall volume to total vessel volume (Normalized wall index) should be used instead of using maximum wall volume [e.g. PMID: 33183741]. This would account for individual variation in overall vessel size.

We thank the Reviewer for this suggestion and reference. However, we prefer to use CWV as our outcome because we have previously validated it in CAHHM where we’ve published CWV ranges related to cardiovascular risk; which is useful clinically [Anand SS, Tu JV, Desai D, et al. Cardiovascular risk scoring and magnetic resonance imaging detected subclinical cerebrovascular disease. Eur Heart J - Cardiovasc Imaging. 2020;21(6):692-700]. Of note, in the sensitivity analysis stratified by sex to account for sex variation, all results were consistent between men and women.

We have now added this sentence to our Discussion (this also addresses the Reviewer’s comment below on clinically relevant cut-off values for CWV):

“We have previously shown in CAHHM how simple cardiovascular risk scores were significantly associated with CWV, where mean (SD) CWV for low, medium, and high INTERHEART risk score categories were 881.5 (163.1), 915.4 (166.6), and 940.9 (172.9) mm3, respectively.”

If the Academic Editor still would like to have this additional analysis, we can conduct it.

-Results:

o Information in the text do not match the information in the table, e.g., in the Results - participant characteristics section it says "54.8% of participants were women", but in Table 1 it says 56%. Same section: "92% of cohort postcodes were in urban areas", but in Table 3 it is 96.2%. It seems that information in the text was mixed up with the “British Columbia” column. Authors should double-check tables and text to correct any discrepancies.

We thank the Reviewer for noting this error. This change has been made for % women. As for urbanicity, we wrote that: “For all the regions, over 92% of the cohort’s postal codes were in urban areas.” i.e. that the minimum % for any of the provinces was 92%.

-Discussion

Any clinically relevant cut-off values for CWV? Is the mean value high (900 mm³)? Could be more addressed in the discussion and may help to contextualize the results and health of study participants to previous studies.

We thank the Reviewer for this valuable suggestion which also reinforces our choice of using CWV as our outcome measure. As noted above we have added the following sentence to the discussion: “We have previously shown in CAHHM that simple cardiovascular risk scores were significantly associated with CWV, where mean (SD) CWV for low, medium, and high INTERHEART risk score categories were 881.5 (163.1), 915.4 (166.6), and 940.9 (172.9) mm3, respectively. Therefore, the overall mean in the current analysis set (900 mm3) corresponds to someone with a low-moderate IHRS.”

I agree with Reviewer 2 that the study was rather cross-sectional in nature than longitudinal. The authors argue “Exposure to air pollution values represented a time frame prior to knowledge of the outcome for each participant, i.e., the air pollution data collected for the 5-year period prior to the MRI. We are therefore comfortable describing this as a prospective association, despite the lack of longitudinal follow-up.” However, since no information exists on the CVW thickness before the exposure window, one cannot argue that the thickness was not present at this exposure window or even before. Therefore, I suggest revising this sentence and to name it a cross-sectional association.

We acknowledge that the study design is not straight-forward because CAHHM itself is a prospective cohort study and the air pollution exposure measures are in the years prior to the outcome measurement but recognize that these air pollution estimates are rather stable over time in the years to follow as well (as noted in the discussion). Thus, to adopt a more conservative approach and because of the lack of longitudinal follow-up, we have now changed the study design to cross-sectional analysis in the title and abstract and have removed this sentence from the discussion.

Optional: instead of A, B, C, in the plots, naming the air pollutants would improve the readability of the plots.

We thank the Reviewer for this optional comment. We prefer to leave as A, B, C, to avoid confusion in case of naming of the air pollutants in Figure 2 where we are testing the interactions for each pollutant within the other 2 pollutant tertiles.

We thank the Academic Editor and external Reviewer for their feedback and hope that our revised manuscript is now deemed acceptable for publication in PLOS One.

---

## [Decision Letter · Decision Letter 2]

30 May 2024

PONE-D-23-23654R2Exposure to Air Pollutants and Subclinical Carotid Atherosclerosis Measured by Magnetic Resonance Imaging: A cross-sectional analysisPLOS ONE

Dear Dr. de Souza,

Thank you for submitting your manuscript to PLOS ONE. After careful consideration, we feel that it has merit but does not fully meet PLOS ONE’s publication criteria as it currently stands. Therefore, we invite you to submit a revised version of the manuscript that addresses the points raised during the review process.

**ACADEMIC EDITOR: **The authors have addressed many changes as advised and have improved the article. The article may be accepted after these last few recommendations. Following issue still needs to be addressed:May be accepted after minor revisions:Editor's comment last time: "Tables S4 through S8 appear to include the study's most significant findings. Given that these are the most significant findings, why are these tables listed in the supplementary materials? These should be mentioned in the main manuscript."   The author responded that the tables are summarized in a few figures. I would strongly advise you to include all of those tables (S4 through S8) in the manuscript. These tables are closely related to the manuscript's actual objective.    Please also consider the reviewer's remarks and final checks for proofreading issues.

We look forward to receiving your revised manuscript.

Kind regards,

Muhammad Maaz Arif, M.B.B.S, M.Phil

Academic Editor

PLOS ONE

Journal Requirements:

Additional Editor Comments:

May be accepted after minor revisions.

Reviewers' comments:

Reviewer's Responses to Questions

**Comments to the Author**

1. If the authors have adequately addressed your comments raised in a previous round of review and you feel that this manuscript is now acceptable for publication, you may indicate that here to bypass the “Comments to the Author” section, enter your conflict of interest statement in the “Confidential to Editor” section, and submit your "Accept" recommendation.

Reviewer #4: (No Response)

2. Is the manuscript technically sound, and do the data support the conclusions?

Reviewer #4: Yes

3. Has the statistical analysis been performed appropriately and rigorously? 

Reviewer #4: Yes

4. Have the authors made all data underlying the findings in their manuscript fully available?

Reviewer #4: Yes

5. Is the manuscript presented in an intelligible fashion and written in standard English?

Reviewer #4: Yes

6. Review Comments to the Author

Reviewer #4: This article was well performed and suggested to be published soon, while there were some additional information should be disclosed:

1. the sensitive of Subclinical Carotid Atherosclerosis Measured by Magnetic Resonance Imaging.How is it varied by time? it is an important basis for the observation period.

2.the difinition of Carotid artery vessel wall volume should be described more minutely.

7. PLOS authors have the option to publish the peer review history of their article (what does this mean?). If published, this will include your full peer review and any attached files.

Reviewer #4: **Yes: **Minjin Peng, Taihe Hospital, Hubei University of Medicine, https://orcid.org/0000-0002-1350-4780

---

## [Author Response · Author response to Decision Letter 2]

4 Jun 2024

ACADEMIC EDITOR Comments:

The authors have addressed many changes as advised and have improved the article. The article may be accepted after these last few recommendations. 

We thank the Academic Editor for this comment and hope the article is deemed to be accepted for publications now that we have addressed the last few recommendations.

Following issue still needs to be addressed:

May be accepted after minor revisions:

Editor's comment last time: "Tables S4 through S8 appear to include the study's most significant findings. Given that these are the most significant findings, why are these tables listed in the supplementary materials? These should be mentioned in the main manuscript." The author responded that the tables are summarized in a few figures. I would strongly advise you to include all of those tables (S4 through S8) in the manuscript. These tables are closely related to the manuscript's actual objective. 

We thank the Academic Editor for this suggestion, and we have now accordingly moved tables S4-S8 to the main manuscript.

Please also consider the reviewer's remarks and final checks for proofreading issues.

We thank the Academic Editor for this comment and have now proofread and revised the article accordingly.

4. The manuscript may be accepted after minor revisions.

We thank the Academic Editor for this comment and hope the article is deemed to be accepted for publications now that we have addressed the last few recommendations.

Reviewer #4 Comments:

This article was well performed and suggested to be published soon, while there were some additional information should be disclosed:

We thank the Reviewer for echoing the strengths and comprehensive analyses conducted within our study.

1. the sensitive of Subclinical Carotid Atherosclerosis Measured by Magnetic Resonance Imaging.

How is it varied by time? it is an important basis for the observation period.

We have now added the following section to the discussion [page 23, line 1]: 

“Compared to cIMT, CWV includes the adventitia, the source of vasa vasorum.12 In terms of sensitivity of MRI-measured CWV, studies have suggested adventitial thickening to be an early sign of atherosclerosis, whereas a dense network of adventitial vasa vasorum can signify progression of atherosclerosis to symptomatic disease.12 “

2.the difinition of Carotid artery vessel wall volume should be described more minutely.

We have now added a detailed description under methods [page 8, under subclinical MRI outcome]: 

“The lumen was defined semi-automatically from axial bright blood images of the time of flight sequence which were reconstructed at 2 mm intervals. The outer wall of the carotid artery was semi-automatically defined and adjusted as needed by expert readers. The area of the vessel wall in each image was estimated by subtracting the lumen area from the outer wall vessel area. Vessel wall volume per slice was calculated by multiplying by 2 mm per slice. Vessel wall volumes for right and left carotid arteries were estimated by integrating the volume for the total number of slices for each artery.”

We thank the Academic Editor and external Reviewer for their feedback and hope that our revised manuscript is now deemed acceptable for publication in PLOS One.

---

## [Editor Report · Decision Letter 3]

21 Aug 2024

Exposure to Air Pollutants and Subclinical Carotid Atherosclerosis Measured by Magnetic Resonance Imaging: A cross-sectional analysis

PONE-D-23-23654R3

Dear Dr. de Souza,

We’re pleased to inform you that your manuscript has been judged scientifically suitable for publication and will be formally accepted for publication once it meets all outstanding technical requirements.

Kind regards,

Muhammad Maaz Arif, M.B.B.S, M.Phil

Academic Editor

PLOS ONE

Additional Editor Comments (optional):

Accept. Manuscript should be thoroughly checked for proofreading errors before the final galley proof.

---

## [Editor Report · Acceptance letter]

2 Sep 2024

PONE-D-23-23654R3 

PLOS ONE

Dear Dr. de Souza, 

I'm pleased to inform you that your manuscript has been deemed suitable for publication in PLOS ONE. Congratulations! Your manuscript is now being handed over to our production team.

Kind regards, 

on behalf of

Dr. Muhammad Maaz Arif 

Academic Editor

PLOS ONE